# Language Models with Image Descriptors are Strong Few-Shot Video-Language Learners

**Zhenhailong Wang**[1][*], **Manling Li**[1][*], **Ruochen Xu**[2], **Luowei Zhou**[2][†],
**Jie Lei**[3], **Xudong Lin**[4], **Shuohang Wang**[2], **Ziyi Yang**[2], **Chenguang Zhu**[2],
**Derek Hoiem**[1], **Shih-Fu Chang**[4], **Mohit Bansal**[3], **Heng Ji**[1]
[1]UIUC  [2]MSR  [3]UNC  [4]Columbia University
{wangz3,hengji}@illinois.edu

## Abstract

The goal of this work is to build flexible video-language models that can generalize to various video-to-text tasks from few examples. Existing few-shot video-language learners focus exclusively on the encoder, resulting in the absence of a video-to-text decoder to handle generative tasks. Video captioners have been pretrained on large-scale video-language datasets, but they rely heavily on finetuning and lack the ability to generate text for unseen tasks in a few-shot setting. We propose **VidIL**, a few-shot **Vid**eo-language Learner via **I**mage and **L**anguage models, which demonstrates strong performance on few-shot video-to-text tasks without the necessity of pretraining or finetuning on any video datasets. We use image-language models to translate the video content into frame captions, object, attribute, and event phrases, and compose them into a temporal-aware template. We then instruct a language model, with a prompt containing a few in-context examples, to generate a target output from the composed content. The flexibility of prompting allows the model to capture any form of text input, such as automatic speech recognition (ASR) transcripts. Our experiments demonstrate the power of language models in understanding videos on a wide variety of video-language tasks, including video captioning, video question answering, video caption retrieval, and video future event prediction. Especially, on video future event prediction, our few-shot model significantly outperforms state-of-the-art supervised models trained on large-scale video datasets. Code and processed data are publicly available for research purposes at https://github.com/MikeWangWZHL/VidIL.

## 1 Introduction

One major gap between artificial intelligence and human intelligence lies in their abilities to generalize and perform well on new tasks with limited annotations. Recent advances in large-scale pre-trained generative language models [45, 6, 71, 24] have shown promising few-shot capabilities [72, 43, 63] in understanding natural language. However, few-shot video-language understanding is still in its infancy. A particular limitation of most recent video-language frameworks [28, 21, 61, 68, 67, 25, 64, 34] is that they are encoder-only, which means they do not have the ability to generate text from videos for purposes such as captioning [62, 57], question answering [60], and future prediction [23]. Meanwhile, unified video-language models [36, 49] that are capable of language decoding still rely heavily on finetuning using a large number of manually annotated video-text pairs, therefore cannot adapt quickly to unseen tasks. Few-shot video-to-text decoding is challenging because the natural language supervision for learning video-language representation is typically based on subtitles and automatic

---

[*]Equal contribution.
[†]Currently at Google Brain.

36th Conference on Neural Information Processing Systems (NeurIPS 2022).

speech recognition (ASR) transcripts [39, 68], which differ significantly from downstream tasks in terms of distribution and may have poor semantic alignment across vision and text modalities.

We propose to address this problem by harnessing the few-shot power of frozen large-scale language models, such as InstructGPT [40]. Our inspiration is derived from the fact that humans are excellent visual storytellers [15], with the ability to piece together a coherent story from a few isolated images. To mimic this, we propose **VidIL**, a few-shot **Vid**eo-language Learner via **I**mage and **L**anguage models, to use image models to provide information about the visual content in the video (as well as optionally use ASR to represent speech), and then we instruct language models to generate a video-based summary, answer, or other target output for diverse video-language tasks.

The main challenge of understanding videos is that, videos contain rich semantics and temporal content at multiple granularities. Unlike static images which depict objects, attributes and events in a snapshot, the temporal dimension of videos further conveys the state changes of the objects, actions, and events. For example, in Figure 1, the individual frame captions of the video clip only describe static visual features such as *"a person holding a green object in hand"*. In contrast, a correct video-level description would be *"a woman makes realistic looking leaves and flowers for a cake"*, which involves reasoning over a collection of objects and events that occur at different timestamps in the video clip, such as *"cake decorating"* and *"flowered design"*. Hence, to inform video-level description and queries, we need to represent all of this information and its temporal ordering.

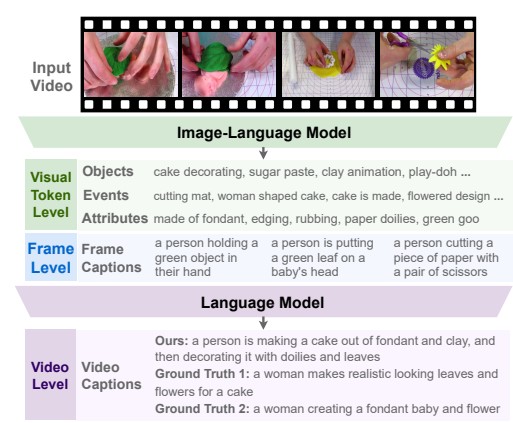

Figure 1: Multiple levels of information in videos.

To address the unique challenges of videos, we propose to decompose a video into three levels: the video output, frame captions, and visual tokens (including objects, events, attributes). One major benefit from this hierarchical video representation is that we can separate the visual and temporal dimensions of a video. We leverage frozen image-language foundational models at lower levels to collect salient visual features from the sparsely sampled frames. Specifically, we first leverage a pretrained image-language contrastive model CLIP [44] to perform visual tokenization, based on the similarity score between frames and tokens of objects, events and attributes. The tokenization is done under the guidance of semantics role labeling [14], which provides us with candidate events with involved objects and related attributes. Next, in order to capture the overall semantics at the frame level, we employ the pretrained image captioner in the image-language model BLIP [26] to obtain frame captions. Finally, we instruct a pretrained large language model using in-context learning [40, 13, 51, 48] to interpret visual tokens and frame captions into the target textual output. In detail, we temporally order visual tokens and frame captions using specially designed prompts such as *"First...Then...Finally"*, to instruct the pretrained language model to track the changes of objects, events, attributes and frame semantics along the temporal dimension.

**Without pretraining or finetuning on any video datasets**, we show that our approach outperforms both video-language and image-language state-of-the-art baselines on few-shot video captioning and question answering tasks. Moreover, on video-language event prediction, our approach significantly outperforms fully-supervised models while using only 10 labeled examples. We further demonstrate that our generative model can benefit broader video-language understanding tasks, such as text-video retrieval, via pseudo label generation. Additionally, we show that our model is highly flexible in adding new modalities, such as ASR transcripts.

## 2 Related Work

### 2.1 Image-Language Models and Their Applications on Video-Language Tasks

Large-scale image-language pretraining models optimize image-text matching through contrastive learning [44, 17] and multimodal fusion [65, 27, 58, 66, 35, 52, 8, 29, 73, 70, 18, 16]. Recently,

BLIP [26] proposes a bootstrapping image-language pretraining framework with a captioner and a filterer which has shown promising performance on various image-language tasks. However, video-language pretraining [25, 36, 28, 38, 3, 1, 42, 33] is still hindered by noisy and domain-specific video datasets [74, 22, 39]. Naturally, researchers start to explore transferring the rich knowledge from image models to videos. Different from the traditional way of representing videos by 3D dense features [12], recent work [21, 25] proves that sparse sampling is an effective way to represent videos, which facilitates applying pre-trained image-language models to video-language tasks [37, 11]. Specifically, the image-language model BLIP [26] sets new state-of-the-art on zero-shot retrieval-style video-language tasks, such as video retrieval and video question answering. However, for generation-style tasks such as domain-specific video captioning, video-language model UniVL [36] still leads the performance but highly rely on fine-tuning. In this work, we extend the idea of leveraging image-language models to a wide variety of video-to-text generation tasks. We further connect image-language models with language models which empowers our model with strong generalization ability. We show that the knowledge from both image-language pretraining and language-only pretraining can benefit video-language understanding in various aspects.

## 2.2 Unifying MultiModal Tasks with Language Models

The community has paid much attention to connecting different modalities with a unified representation recently. Text-only generation models, such as T5 [46], have been extended to vision-language tasks by text generation conditioned on visual features [9, 53, 50, 75, 55]. In order to fully leverage the generalization power from pretained language models, [63] represents images using text in a fully symbolic way. [32] includes more modalities such as video and audio, but requires annotated video-text data to jointly training the language model with the video and audio tokenizer. In this work, we propose a temporal-aware hierarchical representation for describing a video textually. To our knowledge, we are the first work to leverage prompting a frozen language model for tackling few-shot video-language tasks with a unified textual representation. Concurrent work Socratic [69] uses a zero-shot language-based world-state history to represent long videos with given time stamps, while our model can quickly adapt to different video and text distributions with few examples. Furthermore, we show that by injecting *temporal markers* to the prompt we can make a pre-trained language model understand fine-grained temporal dynamics in video events. Compared with the concurrent work Flamingo [2], which requires dedicated vision-language post-pretraining, our framework does not require to pretrain or finetune on any video data. Our framework is simple and highly modulated where all the components are publicly available. Additionally, our framework is more flexible on adding new modalities, e.g., automatic speech recognition, without the need for complex redesigning.

## 3 Method

We propose a hierarchical video representation framework which decomposes a video into three levels, i.e., **visual token level**, **frame level** and **video level**. The motivation is to separate the spatial and temporal dimension of a video in order to leverage image-language and language-only foundation models, such as CLIP [44] and GPT-3 [6]. All three levels use a unified textual representation which enables us to leverage the powerful few-shot ability from pretrained language models.

### 3.1 Frame Level: Image Captioning

Following [21] we first perform sparse sampling to obtain several video frames. Unless otherwise specified, we sample 4 frames for frame level and 8 frames for visual token level. We then feed each frame into a pre-trained image-language model to obtain frame level captions. An example can be found in the blue part of Figure 2. In our experiments, we use BLIP [26], a recent image-language framework containing both image-grounded encoder and decoder, for generating frame captions. We follow [26] to do both captioning and filtering on each frame. However, as mentioned in Section 1, videos contain rich semantics and temporal contents at multiple granularities. It is not enough to generate video-level target text such as video captions solely based on frame captions. Thus, we further perform visual tokenization for each frame to capture features at a finer granularity.

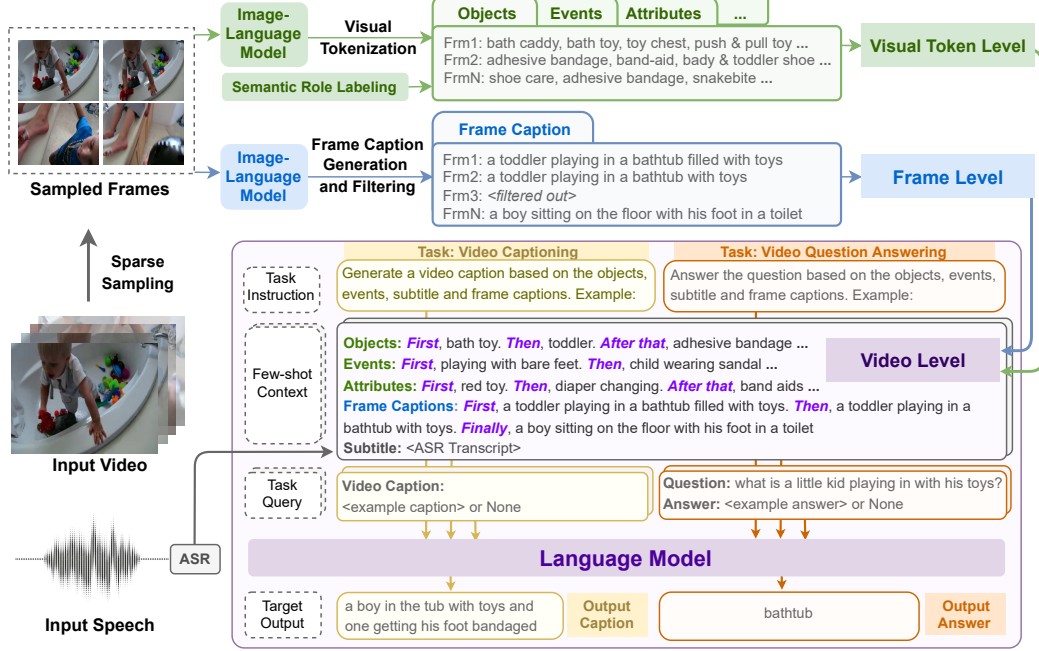

Figure 2: Overview of VidIL framework. We represent a video in a unified textural representation containing three semantic levels: **visual token level**, **frame level**, and **video level**. At visual token level, we extract salient objects, events, attributes for each sampled frame. At frame level, we perform image captioning and filtering. At video level, we construct video representation by aggregating the visual tokens, frame captions and other text modalities such as ASR, using a few-shot temporal-aware prompt. We then feed the prompt to a pre-trained language model together with task-specific instructions to generate target text for a variety of video-language tasks. Examples of the full prompt for different tasks can be found in Appendix **??**.

## 3.2 Visual Token Level: Structure-Aware Visual Tokenization

At this level, we aim to extract the textual representations of salient visual token types, such as objects, events and attributes. We found that pre-defined classes for classification, such as those in ImageNet [10], are far from enough for covering the rich semantics in open-domain videos. Thus, instead of using classification-based methods for visual tokenization as in previous work [32, 63], we adopt a retrieval-based visual tokenization approach by leveraging pre-trained contrastive image-language models. Given a visual token vocabulary which contains all candidate object, event, and attribute text phrases, we compute the image embedding of a frame and the text embeddings of the candidate visual tokens using a contrastive multi-modal encoder, CLIP [44]. We then select top 5 visual tokens per frame based on the cosine similarity of the image and text embeddings. An example of the extracted object tokens can be found in the green part of Figure 2.

Unlike in images where objects and attributes already cover most visual features, events are more informative in videos. In order to discover events from video frames, we construct our own event vocabulary by extracting event structures from Visual Genome [19] synsets[3] using **Semantic Role Labeling**. Specifically, we first select the phrases that contain at least one verb and one argument as events. Then we remove highly similar events based on their sentence similarity using Sentence-BERT [47] embeddings. For object vocabulary, we adopt OpenImage [20] full classes (~20k), instead of using the visually groundable subset (~600) as in concurrent work [69]. We found that using *large but noisy* vocabulary is more effective than using *small but clean* vocabulary in our retrieval-based setting with CLIP. For attribute vocabulary, we adopt visual genome attribute synset. In Section 4.6, we provide ablation study on the impact of different types of visual tokens. The statistics of visual token vocabulary can be found in Appendix Table **??**.

---

[3]We use the keys in Visual Genome [19] object synsets which contains frequent <verb,object> pairs.

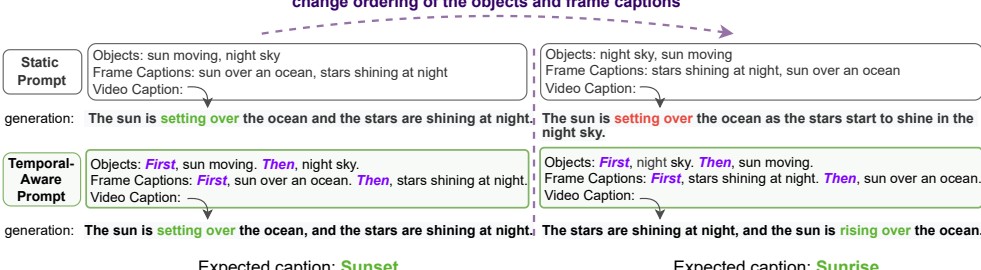

Figure 3: Temporal-aware prompt successfully distinguishes the **Sunset** and **Sunrise** scenarios based on the temporal ordering change of objects and frame captions, while the static prompt fails.

## 3.3 Video Level: Temporal-Aware Few-shot Prompting

Once we obtain the textual representation from frame level and visual token level, the final step is to put the pieces together to generate a video level target text. The goal is to build a model that can be quickly adapted to any video-to-text generation task with only a few examples. To this end, we propose to leverage large-scale pre-trained language models, such as GPT-3 [6], with a temporal-aware few-shot prompt. As shown in Figure 2, our framework can be readily applied to various video-to-text generation tasks, such as video captioning and video question answering, with a shared prompt template. The proposed prompting strategy enables a language model to attend to the lower level visual information as well as taking into account the temporal ordering.

Here, we use the video captioning task depicted in Figure 2 to illustrate the details. The few-shot prompt consists of three parts: **instruction**, **few-shot context**, and **task query**. The **instruction** is a concise description of the generation task, e.g., `"Generate a video caption based on the objects, events, attributes and frame captions. Example:"`, which is proved to be effective in zero-shot and few-shot settings [6, 59]. The **few-shot context** contains the selected in-context examples as well as the test video instance. Each video instance is represented by the aggregated visual tokens[4], e.g., `"Objects: First, bath toy. Then,..."`, the frame captions, such as `"Frame Captions: First, a toddler playing in a bathtub filled with toys. Then,..."`, and the ASR inputs if available, e.g., `"Subtitle:<ASR Transcript>"`. Finally, the **task query** is a task-specific suffix indicating the target text format, e.g. `"Video Caption:"`. For in-context examples (omitted here for simplicity), the task query is followed by ground truth annotation, while for the test instance, the generation starts at the end of the task query.

Formally, we denote the instruction line as $\mathbf{t}$, few-shot context as $\mathbf{c}$, the task query as $\mathbf{q}$, and the target text as $\mathbf{y}$, where $\mathbf{y} = (y_1, y_2, ..., y_L)$. The generation of the next target token $y_l$ can be modeled as:

$$y_l = \arg\max_y p(y|\mathbf{s}, \mathbf{c}, \mathbf{q}, y_{<l}) \tag{1}$$

In order to capture the temporal dynamics between frames and visual tokens, we further propose to inject **temporal markers** to the prompt. As shown in the few-shot context in Figure 2, each visual token and frame caption is prefixed with a natural language phrase indicating its temporal ordering, e.g., `"First,","Then,"`, and `"Finally,"`. We found adding the temporal marker can make the language model conditioned on not only literal but also temporal information of the context. We show an example in Figure 3, where we compare our *temporal-aware prompt* with a *static prompt* on video captioning using InstructGPT. Again, the in-context examples are omitted here, which can be found in Appendix **??**. In this example, the only difference between these two contexts is the ordering of the visual tokens and the frame captions. For the context on the left, where `"sun moving"` appears before `"night sky"`, we are expected to see a story talking about **sunset**, while for the context on the right, we are expected to see **sunrise**. We can see the static prompt generates captions about sunset for both contexts, while the temporal-aware prompt can capture temporal ordering correctly and generate sunrise for the context on the right.

---

[4]To obtain video level visual tokens, the visual tokens extracted from each frame are further ranked and ordered based on frequency and frame index. More details can be found in Appendix **??**.

## 4 Experiments

### 4.1 Experimental Setup

To comprehensively evaluate our model, we show results on four video-language understanding tasks in few-shot settings: video captioning, video question answering (QA), video-language event prediction, and text-video retrieval. We compare our approach with state-of-the-art approaches on five benchmarks, i.e, MSR-VTT [62], MSVD [7], VaTeX [57], YouCook2 [74], and VLEP [23]. The statistics of the datasets can be found in Table 1. For more details please refer to Appendix **??**.

**Implementation Details.** We use CLIP-L/14[6] as our default encoder for visual tokenization. We adopt BLIP captioning checkpoint[7] fine-tuned on COCO [31] for frame captioning. We use InstructGPT [40] as our default language model for generating text conditioned on the few-shot prompt. To construct event vocabulary, we use the semantic role labeling model from AllenNLP[8]. The experiments are conducted on 2 NVIDIA V100 (16GB) GPUs. All few-shot finetuning on baselines and semi-supervised training are performed on 2 Nvidia V100 16G GPUs.

Table 1: Statistics of datasets in our experiments

| Dataset | Task | Split Count # train / # eval |
|---|---|---|
| MSR-VTT [62] | Captioning; QA | 6,513 / 2,990 |
| MSR-VTT [62] | Retrieval | 7,010 / 1,000 |
| MSVD [7] | Question Answering | 30,933 / 13,157 |
| VaTeX v1.1[5] [57] | Captioning; Retrieval | 25,991 / 6,000 |
| YouCook2 [74] | Captioning | 10,337 / 3,492 |
| VLEP [23] | Event Prediction | 20,142 / 4,192 |

**In-context Example Selection.** From our preliminary experiments, we find that the generation performance is sensitive to the quality of in-context examples. For example, for QA tasks such as MSVD-QA where the annotations are automatically generated, the <question, answer> pair in randomly selected in-context examples can be only weakly-correlated with the video context. Thus, instead of using a fixed prompt for each query, we dynamically filter out the irrelevant in-context examples. Specifically, given a randomly sampled *M*-shot support set from the training set, we select a subset of *N*-shots as in-context examples based on their SentenceBERT [47] similarities with text queries. Furthermore, we reorder the selected examples in ascending order based on the similarity score to account for the recency bias [72] in large language models. For QA tasks, we choose the most relevant in-context examples by comparing with questions. While for captioning task, we compare with frame captions. If not otherwise specified, we use *M=10* and *N=5*, which we consider as 10-shot training.

### 4.2 Few-shot Video Captioning

We report BLEU-4 [41], ROUGE-L [30], METEOR [5], and CIDEr [54] scores on three video captioning benchmarks covering both open-domain (MSR-VTT, VaTeX) and domain-specific (YouCook2) videos. We compare with both state-of-the-art video captioner (UniVL [36]) and image captioner (BLIP [26]). In order to implement the BLIP baseline for few-shot video captioning, we extend the approach used for text-video retrieval evaluation in [26] to video-language training. Specifically, we concatenate the visual features of sampled frames and then feed them into the image-grounded text-encoder to compute the language modeling loss. This is equivalent to stitching the sampled frames into a large image and then feeding it to BLIP for image captioning. We found that this simple approach results in very strong baselines.

As shown in Table 2, existing methods have strong bias on certain datasets. For example, UniVL performs well on YouCook2 but fails on MSR-VTT and VaTeX, while BLIP performs the opposite. This is because UniVL is pretrained on HowTo100M which favors instructional videos, i.e., YouCook2, while BLIP is pre-training on image-caption pairs which favors description-style captions, i.e., MSR-VTT and VaTeX. On the contrary, our model performs competitively on both open-domain and instructional videos, and significantly outperforms the baselines on the average CIDEr score across all three benchmarks. This indicates that by leveraging language models, we can maintain strong few-shot ability regardless of the video domain or the target caption distribution.

---

[6]https://huggingface.co/openai/clip-vit-large-patch14
[7]https://github.com/salesforce/BLIP#finetuned-checkpoints
[8]https://docs.allennlp.org/models/main/models/structured_prediction/predictors/srl/

Table 2: 10-shot video captioning results. ♠ indicates concurrent work. The reported *Flamingo* [2] results are using 16 shots. *#Video$_{PT}$* represents the number of videos used for pre-training. *B-4*, *R-L*, *M*, *C* represents *BLEU-4*, *ROUGE-L*, *METEOR* and *CIDEr*. *Avg C* represents the average CIDEr score across all available benchmarks. *ASR* indicates whether the model has access to the ASR subtitles. *BLIP* and *BLIP$_{cap}$* use the pretrained checkpoint and the finetuned checkpoint on COCO captioning. All results are averaged over three random seeds.

| Method | #Video$_{PT}$ | ASR | MSR-VTT Caption | | | | YouCook2 Caption | | | | VaTex Caption | | | | Avg C |
|---|---|---|---|---|---|---|---|---|---|---|---|---|---|---|---|
| | | | B-4 | R-L | M | C | B-4 | R-L | M | C | B-4 | R-L | M | C | |
| *Few-shot* | | | | | | | | | | | | | | | |
| UniVL | 1.2M | No | 2.1 | 22.5 | 9.5 | 3.6 | **3.3** | **25.3** | **11.6** | **34.1** | 1.7 | 15.7 | 8.0 | 2.1 | 13.3 |
| BLIP | 0 | No | **27.7** | 43.0 | 23.0 | **39.5** | 0.7 | 9.0 | 3.4 | 11.5 | 13.5 | 39.5 | 15.4 | 20.7 | 23.9 |
| BLIP$_{cap}$ | 0 | No | 21.6 | 48.0 | 22.7 | 30.2 | 3.7 | 8.6 | 3.8 | 9.4 | 20.7 | 41.5 | 17.4 | 28.9 | 22.8 |
| VidIL(ours) | 0 | No | 26.0 | **51.7** | **24.7** | 36.3 | 2.6 | 22.9 | 9.5 | 27.0 | **22.2** | **43.6** | **20.0** | **36.7** | **33.3** |
| UniVL | 1.2M | Yes | - | - | - | - | 4.3 | 26.4 | 12.2 | 48.6 | 2.7 | 17.7 | 10.2 | 3.4 | 26.0 |
| VidIL(ours) | 0 | Yes | - | - | - | - | **10.7** | **35.9** | **19.4** | **111.6** | **23.2** | **44.2** | **20.6** | **38.9** | **75.3** |
| ♠Flamingo-3B(16) | 27M | No | - | - | - | - | - | - | - | 73.2 | - | - | - | 57.1 | - |
| ♠Flamingo-80B(16) | 27M | No | - | - | - | - | - | - | - | 84.2 | - | - | - | 62.8 | - |
| *Fine-tuning* | | | | | | | | | | | | | | | |
| UniVL | 1.2M | No | 42.0 | 61.0 | 29.0 | 50.1 | 11.2 | 40.1 | 17.6 | 127.0 | 22.8 | 38.6 | 22.3 | 33.4 | 70.2 |
| UniVL | 1.2M | Yes | - | - | - | - | 16.6 | 45.7 | 21.6 | 176.8 | 23.7 | 39.3 | 22.7 | 35.6 | 106.2 |

As discussed in Section 1, video captions describe the content in various semantic levels. The N-gram based metric may not fairly reflect the models' performance in capturing the video-caption alignment. We further verify this hypothesis in Section 4.5. Thus, in addition to automatic metrics, we include qualitative examples illustrated in Figure 4. More examples are in Appendix **??**.

Additionally, for most existing methods and also concurrent work, e.g., Flamingo [2], adding a new modality often requires a dedicated model redesign or retraining. However, the nature of our framework, where we use a unified textual representation for each level, makes it highly flexible for incorporating new modalities. As shown in row 6 in Table, our model can effectively utilize extra information from ASR to obtain significantly better few-shot performance on certain datasets such as YouCook2.

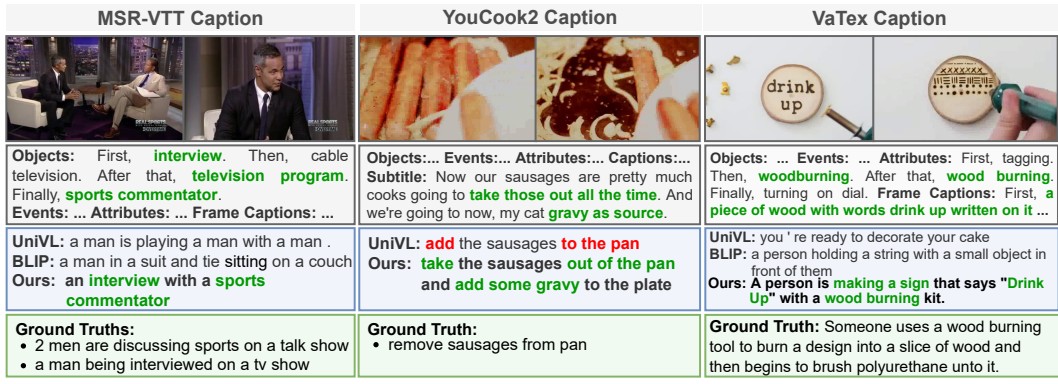

Figure 4: Qualitative examples on video captioning. Grey boxes contain part of the video representation from our model. Blue boxes contain caption generation from different models. Green boxes contain ground truth annotations. **Bold green text** highlights the correct information that is not captured in baseline outputs which can be reasoned from our visual tokens and frame captions.

## 4.3 Few-shot Video Question Answering

We compare the test accuracy of our approach with few-shot pretrained BLIP, BLIP$_{VQA}$ [26], and concurrent work Flamingo [2] on two video question answering benchmarks, MSR-VTT_QA and MSVD_QA. BLIP$_{VQA}$ represents finetuned BLIP on VQA [4] dataset, which is the previous SOTA

on zero/few-shot video question answering. In order to have fairer comparison with $BLIP_{VQA}$, we reduce the shot number to 5 and report the average accuracy on three sets of randomly selected 5-shot examples. As shown in Table 3, our method outperforms previous SOTA by a large margin. Comparing with concurrent work Flamingo, which is post-pretrained on a large number of video-text data, our model is training-free and did not observe any video data. However, with only image-language and language-only knowledge, our 5-shot model is able to outperform 8-shot Flamingo-3B and achieve on-par performance with 4-shot Flamingo-80B.

Table 3: Video QA results. $BLIP_{VQA}$ is finetuned on VQA [4]. ♠ indicates concurrent work. PT, FT indicates pretraining and finetuning.

| Method | #video$_{PT}$ | #video$_{FT}$ | MSR-VTT | MSVD |
|---|---|---|---|---|
| BLIP | 0 | 0-shot | 0.55 | 0.45 |
| BLIP | 0 | 5-shot | 0.84 | 0.53 |
| $BLIP_{VQA}$ [26] | 0 | 0-shot | 19.2 | 35.2 |
| VidIL(ours) | 0 | 5-shot | **21.2** | **39.1** |
| ♠Flamingo-3B [2] | 27M | 4-shot | 14.9 | 33.0 |
| ♠Flamingo-3B [2] | 27M | 8-shot | 19.6 | 37.0 |
| ♠Flamingo-80B [2] | 27M | 4-shot | 23.9 | 41.7 |
| ♠Flamingo-80B [2] | 27M | 8-shot | 27.6 | 45.5 |
| ALPRO [25] | 2M | full-shot | 42.1 | 45.9 |

Table 4: Accuracy (%) on VLEP hidden test set.

| Method | #video$_{FT}$ | Acc |
|---|---|---|
| VLEP [23] | 20142 | 67.5 |
| MERLOT [68] | 20142 | 68.4 |
| VidIL(ours) | 10-shot | **72.0** |
| Human | - | 90.5 |

## 4.4 Few-shot Video-Language Event Prediction

In this section, we show that our model not only can answer questions about the video visual features but also answering "What is more likely to happen next?". Given a video with associated subtitle transcript as premise, the video-language event prediction (VLEP) task is to predict the most likely future event. The original VLEP [23] paper formulates the problem as a binary classification problem where the model will be chosen from two possible future event candidates. Instead, we formulate this problem as another video-to-text generation problem to fit into our framework. Figure 5 depicts an example with the same format as in Figure 2. Similar to the evaluation setting in QA, the generated free-form text will first be mapped to one of the two candidate answers using SentenceBert [47], and then calculate the accuracy. In Table 4, we report accuracy on the hidden test set of VLEP [23]. To our surprise, our 10-shot model outperforms state-of-the-art fully-supervised baseline, i.e., MERLOT [68], by a large margin ($\sim 4\%$). This shows that our model has strong few-shot ability not only on video-language understanding but also on prediction. Since event prediction tasks rely heavily on temporal ordering, we show that with the proposed temporal-aware prompting, language models can be guided to capture temporal dynamics between historical and future events.

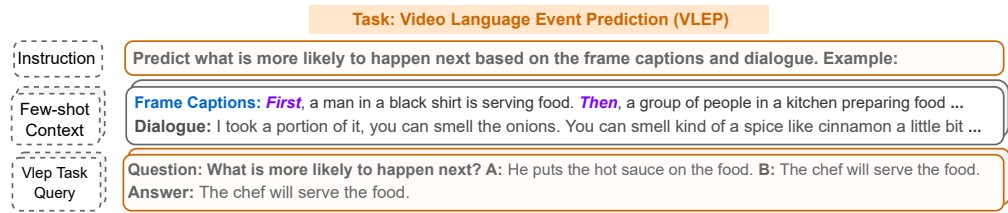

Figure 5: Prompt for VLEP task.

## 4.5 Semi-supervised Text-Video Retrieval

In addition to video-to-text generation tasks, we show that a broader range of video-language tasks can benefit from our few-shot video captioner from a data perspective. Here, we consider a low-budget semi-supervised setting where we only have a few labeled video-caption pairs and a large amount of unlabeled videos. The idea is to leverage our video captioner to generate **pseudo labels** for training any given vision-language models. As a case study, we evaluate on two text-video retrieval benchmarks, i.e., MSR-VTT and VaTeX. We use greedy decoding to generate pseudo caption for each video in the training set. We then train an identical base model, i.e., BLIP, using different pseudo labeled data as well as ground truth annotations. We report Recall @ 1 and 5 for both video-to-text

Table 5: Semi-supervised text-video retrieval with 10 labeled examples. $V_{label}$ or $V_{unlabel}$ are the number of labeled and unlabeled videos, respectively. *t_R1* and *t_R* denote video-to-text Recall@1 and 5. *v_R1* and *v_R5* denote text-to-video Recall@1 and 5.

| Model | Pseudo Label | MSR-VTT Retrieval | | | | | VaTex Retrieval | | | | |
|---|---|---|---|---|---|---|---|---|---|---|---|
| | | $V_{label}/V_{unlabel}$ | t_R1 | t_R5 | v_R1 | v_R5 | $V_{label}/V_{unlabel}$ | t_R1 | t_R5 | v_R1 | v_R5 |
| BLIP | - | - | 33.2 | 57.2 | 40.5 | 62.8 | - | 28.2 | 53.4 | **34.0** | 58.6 |
| BLIP | UniVL | 10 / 7010 | 33.1 | 57.3 | 33.6 | 57.7 | 10 / 22685 | 25.5 | 47.7 | 26.1 | 49.1 |
| BLIP | BLIP | 10 / 7010 | 35.6 | 60.8 | 39.8 | 60.4 | 10 / 22685 | 26.3 | 50.5 | 29.3 | 53.6 |
| BLIP | $BLIP_{cap}$ | 10 / 7010 | 35.3 | 58.0 | 39.1 | 63.3 | 10 / 22685 | 23.9 | 46.8 | 27.5 | 49.7 |
| BLIP | VidIL(ours) | 10 / 7010 | **39.6** | **64.5** | **40.8** | **65.2** | 10 / 22685 | **33.3** | **59.1** | 33.7 | **59.5** |
| BLIP | Ground Truth | 7010 / 0 | 43.6 | 66.2 | 43.1 | 67.2 | 22685 / 0 | 40.1 | 66.4 | 40.1 | 66.6 |
| ALPRO [25] | Ground Truth | 140200 / 0 | 32.0 | 60.6 | 33.9 | 60.7 | - | - | - | - | - |
| DRL [56] | Ground Truth | 180000 / 0 | 54.1 | 77.4 | 52.9 | 78.5 | - | - | - | - | - |

and text-to-video retrieval. Table 5 shows that through training on our pseudo labels, we can achieve significant improvements compared with zero-shot BLIP. We also show that the performance gain is not simply a result of training on more data, since finetuning on the pseudo labels generated by other baselines (UniVL, BLIP) is less effective and can even hurt the performance. Furthermore, on MSR-VTT Recall @ 5 we can even achieve comparable performance against BLIP model finetuned on full ground truth annotations.

Another interesting observation is that, compared with the video captioning results in Table 2, we found that the gain of our model over baselines on text-video retrieval is more visible than on captioning. A key factor in performing well on text-video retrieval tasks is to learn a good video-text multi-modal alignment. This result shows that our pseudo labels capture richer video-text alignment that can benefit the retrieval-style downstream task. The N-gram based generation metrics, e.g., BLEU, may not be able to fully reflect the alignment information, due to the variety of semantic levels in video captions. Furthermore, from a data perspective, our video captioner can be viewed as a data augmentation tool which is capable of generating or augmenting any open-domain video-language pretraining datasets with minimal human effort. As a result, we can potentially improve video-language pretraining by constructing a cleaner and more diverse video-text corpus.

Table 6: Impact of visual tokens and temporal dimension.

| | Video Representation | Avg↑ | Std↓ |
|---|---|---|---|
| | Frame | 39.6 | 3.7 |
| | Frame+Object | 40.3 | 2.9 |
| Visual | Frame+Object+Event | 39.9 | 2.8 |
| Token | Frame+Object+Attibute | **40.9** | 2.9 |
| | Frame+Object+Event+Attribute | 40.8 | **2.4** |
| Temporal | Reduce to one frame | 38.5 | 2.4 |
| | Reverse temporal order | 40.7 | 1.7 |

Table 7: Impact of shot selection. *#ICE* indicates the number of in-context examples in the prompt. Details of in-context example selection are in the Appendix.

| #shot | w/o selection | | | w/ selection | | |
|---|---|---|---|---|---|---|
| | #ICE | Avg↑ | Std↓ | #ICE | Avg↑ | Std↓ |
| 5 | 5 | 38.4 | 2.1 | 5 | 40.4 | 1.2 |
| 10 | 10 | 41.3 | 3.6 | 5 | 40.8 | 2.4 |
| 20 | 20 | 42.6 | 3.3 | 5 | 42.2 | 2.0 |
| 30 | 30 | 40.0 | 2.9 | 5 | 41.1 | 1.9 |

## 4.6 Ablation Studies

We perform comprehensive ablation studies on our few-shot prompt including the impact of different video representation, number of shots and in-context selection. All the ablation results are evaluated on MSVD_QA validation set, and we report the mean and standard deviation of each setting on three sets of randomly sampled shots. For the cases **with** in-context example selection, we further select 5 examples as in-context examples from the sampled shots, while for the cases **without** in-context selection, all shots will be feed into the prompt. In Table 6, we show adding visual tokens consistently improves not only the model accuracy but also the model variance. A lower standard deviation indicates that the model is less sensitive to the few-shot sampling.

To further demonstrate the impact of the additional temporal dimension of videos, we perform two ablations on the "Frame+Object+Event+Attribute" setting. First, we reduce the number of

frame captions and visual tokens to be one[9] for each video. We found that the performance drops significantly compared with using the default four frames, which indicates the model's ability to incorporate information from multiple timestamps. Further, we found that fine-grained temporal modeling is rarely required for performing well on current video-language benchmarks. As shown in the ablation result where we reverse the order of all visual tokens and frame captions, the performance decreased only marginally, which indicates that current benchmarks may not be sufficient in reflecting the benefits from better temporal ordering.

In Table 7, we first show that, with the same context length, namely, 5 in-context examples, in-context example selection significantly increases the performance as well as the robustness. At 10-shot, and 20-shot, directly fitting more shots into the prompt results in better performance. In-context selection achieves slightly lower performance but with significantly better efficiency due to shorter context. Interestingly, at 30-shot, in-context selection with 5 examples outperforms directly adding all 30 shots into the prompt. This is showing that in-context selection can help the model utilize a larger number noisy video examples. Nevertheless, we still observe that the benefit of adding more shots saturated at around 20 to 30 shots, even if with in-context selection. we view this as a remaining challenging on how to make language models benefit from longer contexts.

## 5   Conclusions, Limitations and Future Work

This paper proposes VidIL, a few-shot **Vid**eo-language Learner via **I**mage and **L**anguage models. It demonstrates the strong ability of large-scale language models on performing video-to-text tasks when frame features are provided as unified text representations using image-language models. We propose a temporal order aware prompt by decomposing videos into a hierarchical structure, which is able to plug in multiple levels of frame features, along with speech transcripts. Without pretraining on videos, our model outperforms vision-language models learned from large-scale video datasets on a variety of few-shot tasks, such as domain-specific captioning, question answering, and future event prediction. One limitation of using unified textual representation is that we might lose low-level visual features which can be essential for some specific tasks, such as fine-grained spatial visual question answering. We also observe that current video-language benchmarks rarely require explicit temporal tracking on the frames and visual tokens. Future work will focus on leveraging large-scale language models for learning script knowledge from long videos where temporal dynamics are better emphasized.

## 6   Broader Impact

An open-domain few-shot video-language learner has a wide range of beneficial applications for society, such as automatically detecting violent or mature content in videos and helping people with vision impairment understand videos. However, since the language model is pretrained on massive internet-scale text data, there might be unexpected output that can have potential negative impact on the society, such as bias against people of a certain gender, race or sexuality. Future work and dedicated collaboration from the community are needed to alleviate the potential negative societal impact of large language models.

## Acknowledgements

We thank the anonymous reviewers helpful suggestions. This research is based upon work supported in part by U.S. DARPA AIDA Program No. FA8750-18-2-0014 and U.S. DARPA KAIROS Program Nos. FA8750-19-2-1004. The views and conclusions contained herein are those of the authors and should not be interpreted as necessarily representing the official policies, either expressed or implied, of DARPA, or the U.S. Government. The U.S. Government is authorized to reproduce and distribute reprints for governmental purposes notwithstanding any copyright annotation therein.

---

[9]we use the frame caption and visual tokens from the middle frame.

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
