# Appendix: Language Models with Image Descriptors are Strong Few-Shot Video-Language Learners

**Zhenhailong Wang**[1][*], **Manling Li**[1][*], **Ruochen Xu**[2], **Luowei Zhou**[2][†],
**Jie Lei**[3], **Xudong Lin**[4], **Shuohang Wang**[2], **Ziyi Yang**[2], **Chenguang Zhu**[2],
**Derek Hoiem**[1], **Shih-Fu Chang**[4], **Mohit Bansal**[3], **Heng Ji**[1]
[1]UIUC  [2]MSR  [3]UNC  [4]Columbia University
{wangz3,hengji}@illinois.edu

## 1 Additional Qualitative Examples

Additional qualitative examples on MSR-VTT, YouCook2 and VaTex captioning can be found in Figure 1,2. We show that our framework can capture important video semantics (shown in **bold green text**), such as objects, events and attributes, that are missing in the captions generated by baselines.

## 2 Few-shot Prompt Examples

We show a full view of the few-shot prompts used in video captioning (Figure 3, 4), video question answering (Figure 5) and video-language event prediction (Figure 6). Additionally, in Figure 7, we show the omitted in-context examples for Figure 3 in the main body.

## 3 Additional Experimental Details

**Datasets.** For MSR-VTT captioning and question answering, we use the original split containing 6,513 videos for training and 2,990 for testing. For MSR-VTT retrieval, we use the split containing 7,010 videos for training and 1,000 for testing following previous work. For MSVD question answering, we use the original split containing 30,933 questions for training and 13,157 questions for testing. For VaTeX captioning and retrieval, we use the latest v1.1 version[3], which contains 25,991 videos for training and 6,000 videos for public testing. For YouCook2 captioning, we use 10,337 short clips for training and 3,492 for validation following the VALUE [4] benchmark. For Video-Language Event Prediction (VLEP), we report result on the hidden test set using its official CodaLab evaluation server.[4]

Table 1: Statistics of visual token vocabulary.

| Visual Token | Source | Original Size | Final Size |
|---|---|---|---|
| Objects | OpenImage v6 Classe Names | 19,975 | 19,965 |
| Events | Visual Genome Object Synset (keys) | 40,154 | 7,414 |
| Attributes | Visual Genome Attribute Synset (keys) | 18,720 | 16,693 |

---

[*]Equal contribution.

[†]Currently at Google Brain.

[3]Previous work only reports results on 1,500 validation videos, since previous version of VaTeX does not have public testing set.

[4]https://github.com/jayleicn/VideoLanguageFuturePred/tree/main/standalone_eval

36th Conference on Neural Information Processing Systems (NeurIPS 2022).

**Statistics of Visual Token Vocabulary.** We construct our visual token vocabulary based on Open-Image [2] v6 class names[5], visual genome [1] object synsets[6] and visual genome attribute synsets[7]. The statistics can be found in Table 1. Visual genome synsets are <key, value> pairs, where the keys are noisy natural language phrases and the values are the mapped WordNet synsets [6]. For object vocabulary, we perform minimum cleaning by removing fictional character names such as `"robin (fictional character)"`, which we found are highly biased by the CLIP [7] model on video frames. For attribute vocabulary, we clean up attribute synset keys by removing phrases with a cosine similarity larger than 0.9 using SentenceBert [8] embedding, such as `"facing upward"` and `"facing upwards"`. For event vocabulary, we select phrases containing <verb,object> structures from the object synset keys by running semantic role labeling[8]. We then remove semantically similar entries with a threshold of 0.9 based on SentenceBert embeddings.

**Implementation Details for Visual Token Aggregation.** Once we obtained top 5 visual tokens for each frame, we further aggregate them to construct the video-level visual tokens which will be part of the few-shot prompt. We first rank the visual tokens based on their single frame ranking score with the appearance frequency across all frames as tie breaker. In our implementation, we consider up to top 4 video-level visual tokens, we then filter out any visual token that has not been ranked within top 2 in any frames. To identify the ordering of the obtained video-level visual tokens, we consider the frame index from which they are extracted from as their temporal indicator. If a visual token occurs in multiple frames, we use the averaged frame index as its temporal indicator. Finally, in order to apply temporal prompt template to variable number of visual tokens, we use a dynamic template which changes according to the number of tokens. For example, if we have three visual tokens, we remove `"After that"` and only use `"First"`, `"Then"`, `"Finally"`. If we have more then four visual tokens, we repeat `"Then"` or `"After that"` for tokens in the middle.

**Implementation Details for Few-shot Video Captioning Baselines.** In order to finetune the pretrained baselines (UniVL [5], BLIP [3], $\text{BLIP}_{cap}$ [3]) with few annotated examples on video captioning, we set the learning rate to be small and the warm-up steps to be high. Specifically, for UniVL, we set the number of epochs to be $50$ and the linear warmup steps to be $40$. We use a learning rate of $1e$-6 for captioning task without ASR input and $3e$-6 with ASR input. For BLIP and $\text{BLIP}_{cap}$, we set the number of epoches to be 5 with a learning rate of $5e$-7. For each video, we sample 4 frames (each with a size of 224) at training time and 8 frames at test time. We set all batch size to be the same as the few-shot number, i.e., $10$.

**Implementation Details for Semi-supervised Text-Video Retrieval.** We use pretrained BLIP with Vit-B/16[9] as our base model for training on different pseudo labeled datasets as well as ground truth annotations for text-video retrieval. We train the model for one epoch using a batch size of 16 and a learning rate of $5e$-6. For each video, we sample 4 frames (each with a size of 224) at training time and 8 frames at test time. We follow [3] to first select $k$ candidates based on the video-text feature similarity, where the video features are represented by concatenated frame features. We then rerank the selected candidates based on their pairwise Image-Text Matching (ITM) score. We set $k = 64$ for both MSR-VTT and VaTex retrieval.

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

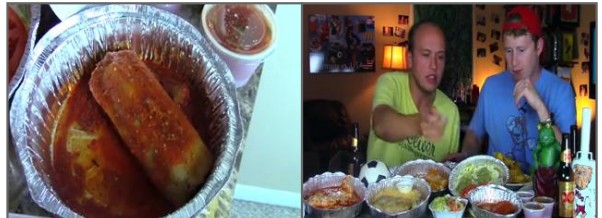

**Objects:** First, cannelloni. Then, enchilada. After that, competitive eating. Finally, **tex-mex food**.
**Events:** First, rolled up sleeve. Then, wrapped items. After that, man eats with hands. Finally, men eating.
**Attributes:** First, black with red sauce. Then, meatfilled. After that, feasting. Finally, holding left overs.
**Frame Captions:** First, a couple of men sitting at a table with bowls of food. Then, **a table topped with lots of food and condiments**. After that, a couple of men sitting at a table with food...

**UniVL:** it ' s got a lot of flavor in it it ' s got a lot of flavor..
**BLIP:** a man and a woman eating food
**BLIPcap:** a couple of men standing in front of a table filled with food
**Ours: two men eat a hearty meal of tex-mex food**

**Ground Truths:**
- two men discuss mexican street food
- two people are sitting in front of a lot of food and talking about it

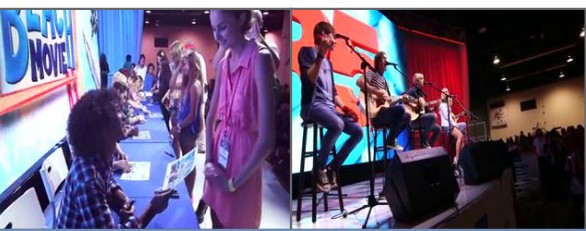

**Objects:** First, **autograph**. Then, afro. After that, **fan convention**. Finally, band-aid.
**Events:** First, hat the girl is wear. Then, child touching. After that, surfers blonde hair. Finally, fans sitting.
**Attributes:** First, **piece signing**. Then, fans. After that, acoustical. Finally, **sing it loud**.
**Frame Captions:** First, a group of people standing around a blue table. Then, a group of people on a stage with microphones. Finally, a group of people sitting on stools on top of a stage.

**UniVL:** a woman ' s wedding ceremony
**BLIP:** a group of people on stage
**BLIPcap:** a group of people standing on top of a stage
**Ours: a band is signing autographs for their fans**

**Ground Truths:**
- a boy band performs and signs autographs
- a band meeting fans and then performing

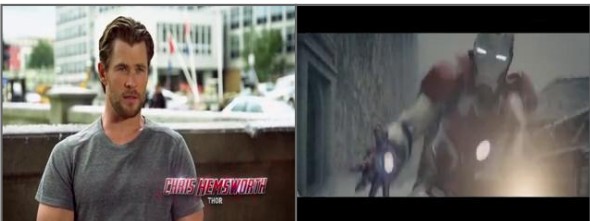

**Objects:** First, **thor**. Then, **avengers**. After that, **ultron**. Finally, **iron man**.
**Events:** First, cap is red. Then, cap is. After that, there is a statue. Finally, cap is black.
**Attributes:** First, wearing red shirt. Then, bending his head. After that, cap. Finally, iron.
**Frame Captions:** First, a man standing in front of a tall building. Then, a scene from the **movie** iron man. Finally, a man with long hair standing in front of a window.

**UniVL:** a man is playing a man is playing a man ' s game
**BLIP:** a man with long hair standing in front of a building
**BLIPcap:** a man standing in front of a tall building
**Ours: a man is watching a movie**

**Ground Truths:**
- a man is being interviewed about a movie
- a video about avengers
- chris helmsworth discusses avengers age of ultron

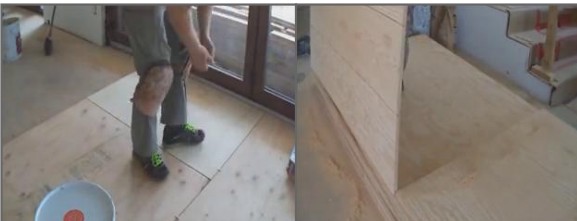

**Objects:** First, step cutting. Then, laminate flooring. After that, wood flooring. Finally, plywood.
**Events:** First, floor shows. Then, **floor trim**. After that, **leveled floors**. Finally, man wearing knee pad.
**Attributes:** First, **covering floor**. Then, **sanding**. After that, push to walk. Finally, stabilizing.
**Frame Captions:** First, a person standing on a hard wood floor. Then, a person sitting on a couch in front of a sliding glass door. Finally, a man standing on top of a hard wood floor.

**UniVL:** the first step is to remove the flooring from the floor joist...
**BLIP:** a person standing on a wooden floor
**BLIPcap:** a person standing in front of a wooden door
**Ours: a man is refinishing a hardwood floor**

**Ground Truths:**
- a man is installing new flooring
- a carpenter places down some wood floring
- a man is fixing the floor

**Objects:** First, storage chest. Then, ranged weapon. After that, **minecraft**. Finally, meat chop.
**Events:** First, block the light. Then, mirrored doors. After that, there is a kitchen. Finally, there are 4 vanilla.
**Attributes:** First, forgotten. Then, dont cross. After that, breaking on left. Finally, clipped in.
**Frame Captions:** First, **a computer generated image of a room in minecraft**. Then, a computer generated image of a stair case. Finally, **a computer screen shot of a room in minecraft**.

**UniVL:** a man is playing a game
**BLIP:** a room in minecraft
**BLIPcap:** a computer generated image of a bathroom with a toilet
**Ours: a man is playing a video game**

**Ground Truths:**
- a man is playing a video game
- a boy eats chicken in minecraft
- gameplay footage of minecraft

Figure 1: Additional qualitative examples on MSR-VTT Captioning.

**YouCook2 Caption**

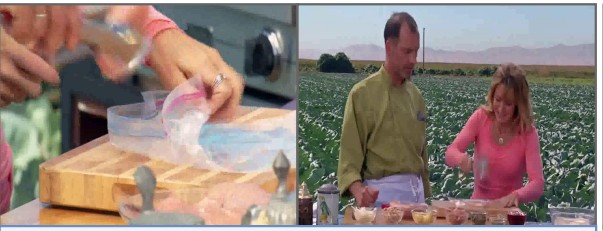

**Objects:** First, **pork loin**. Then, vacuum sealer. After that, plastic wrap. Finally, cooking show.
**Events:** First, person cutting. Then, grey serving tray. After that, cutting board. Finally, dish he is preparing.
**Attributes:** First, cutting food. Then, farmers. After that, removing food. Finally, chopped.
**Frame Captions:** First, a woman in a pink shirt talking to someone ... Finally, a man and a woman preparing food in a field.
**Subtitle:** You going to **pound** it? We want to **give it a nice whack**. It be like I'm beating that hit it hit it Laura, OK? I think we can feel it now.

**UniVL:** pound the chicken breast in the pan
**Ours:** **pound** the **pork loin**

**Ground Truths:**
add a piece of pork in a ziplock bag and pound it

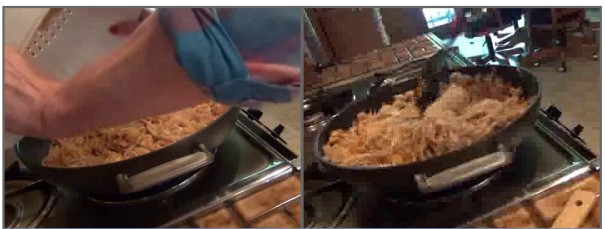

**Objects:** First, deep frying. Then, fried food. After that, food warmer. Finally, fried noodles.
**Events:** First, fried rice. Then, people are eating. After that, fried food. Finally, frying rack.
**Attributes:** First, **stirfried**. Then, non stick. After that, being cooked. Finally, full of food.
**Frame Captions:** First, a **pan** of food on a stove top. Then, a **pan** filled with food on top of a stove. Finally, a person putting food into a **pan** on top of a stove.
**Subtitle:** And **already cooked**. My Udham noodles. Not like this. Oh, I wish I had smellovision This is smelling so good.

**UniVL:** add some seasoning spice udham noodles and mix in
**Ours:** add in the **already cooked** noodles to the **pan**

**Ground Truths:**
add udon noodles to the pan and stirt

**Vatex Caption**

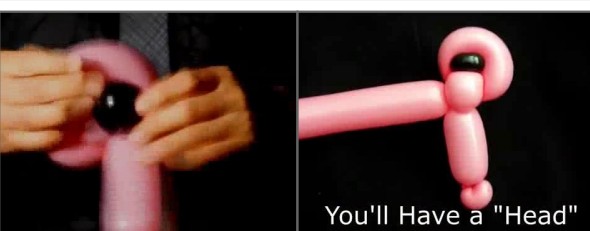

**Objects:** First, bubble blowing toy. Then, human head. After that, **balloon**. Finally, dog toy.
**Events:** First, **floating balloon**. Then, head turned. After that, head looking. Finally, **blow up ornament**.
**Attributes:** First, **helium filled**. Then, head on. After that, head. Finally, rubbery.
**Frame Captions:** First, **a pink flamingo balloon sitting on top of a table**. Then, a clock tower lit up in the dark. Finally, a blurry image of a person in a body suit.

**UniVL:** a child ' s head is a child ' s head .
**BLIP:** a person holding a pink object
**BLIPcap:** a close up of a person holding a pink object
**Ours:** A person **blows up a balloon** and ties it off.

**Ground Truths:**
- A person is twisting blown up balloons into a head figure and then into an animal figure
- A person shows how to make a balloon dog using balloons

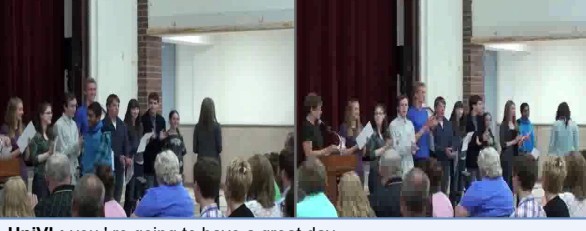

**Objects:** First, spelling bee. Then, **applause**. After that, talent show. Finally, helping hand.
**Events:** First, to shake hands. Then, students are looking. After that, people are standing. Finally, band is white.
**Attributes:** First, handing something. Then, highfiving. After that, moving up. Finally, **clapping her hands**.
**Frame Captions:** First, a group of people standing in front of a crowd of people. Then, a group of people standing in front of a crowd. Finally, a group of people standing in front of a podium.

**UniVL:** you ' re going to have a great day .
**BLIP:** a group of people standing in front of a crowd
**BLIPcap:** a group of people that are standing in front of a microphone
**Ours:** **A group of people standing in front of a podium, clapping their hands**.

**Ground Truths:**
- A woman reads off names in front of crowd and young people get a piece of paper, the crowd applauds.

Figure 2: Additional qualitative examples on YouCook2 and VaTex Captioning.

**Generate a video caption based on the objects, events, attributes and frame captions. Example:**

**Objects:** First, kaval. Then, special agent. After that, detective. Finally, saw u.
**Events:** First, to board the plane. Then, wears blue shirt. After that, bus is hyundai. Finally, tour bus has sup.
**Attributes:** First, coupled. Then, sas. After that, driving away. Finally, expandable.
**Frame Captions:** First, a woman in a car looking out the window. Then, a car with a lot of fire coming out of it. After that, a man laying on top of a bed next to a pair of glasses. Finally, a woman laying in the grass with her eyes closed.
**Video Caption:** it is the clips of a movie

**Objects:** First, animal sports. Then, liger. After that, predation. Finally, lion.
**Events:** First, zebras are playing. Then, zebra is bent over. After that, zebras are eating. Finally, giraffe's find food.
**Attributes:** First, next to zebra. Then, trying to eat. After that, looking for meal. Finally, stalking.
**Frame Captions:** First, a blurry photo of a zebra in a field. Then, a zebra laying on its back in a field. After that, a dog chasing a zebra on the ground. Finally, a lion running through a field with rocks.
**Video Caption:** a lion is catching an zebra

**Objects:** First, western screech owl. Then, eastern screech owl. After that, screech owl. Finally, otter.
**Events:** First, bird has a beak. Then, animated bird. After that, its pupils are green. Finally, eyes are open.
**Attributes:** First, startled. Then, owl shaped. After that, tawny. Finally, used as number.
**Frame Captions:** First, a close up of an owl with an open mouth. Then, a close up of an owl with its mouth open. After that, a close up of an owl with a blurry background. Finally, a blurry picture of a dog laying on the floor.
**Video Caption:** an owl making a weird sound

**Objects:** First, chopper. Then, billfish. After that, cartoon. Finally, colt.
**Events:** First, people look surprize. Then, two kids crouched. After that, animated kid. Finally, man watching blond.
**Attributes:** First, covered with cartoon. Then, gotee. After that, cartoon image. Finally, cartoon.
**Frame Captions:** First, a cartoon picture of a man and a woman talking to each other. Then, a cartoon picture of a person walking next to a cartoon character. After that, a cartoon of a man with a hat on. Finally, a blurry photo of a room with a trash can.
**Video Caption:** cartoon children are confronted by bullies

**Objects:** First, action-adventure game. Then, strategy video game. After that, stage combat. Finally, ac ace.
**Events:** First, player is running. Then, player leans back. After that, two players move. Finally, screen shows game.
**Attributes:** First, suspended on corner. Then, remake. After that, suspended in air. Finally, making the ledge.
**Frame Captions:** First, a screenshot of a video game with a man in a red shirt. Then, a screenshot of a video game with a man on a ledge. After that, a screenshot of a video game with a person on a skateboard. Finally, a screenshot of a video game with a bird in the foreground.
**Video Caption:** this is a video game being played

**Objects:** <query objects>
**Events:** <query events>
**Attributes:** <query attributes>
**Frame Captions:** <query frame captions>
**Video Caption:**

Figure 3: An example of few-shot prompts for MSR-VTT captioning.

**Generate a video caption based on the objects, events, attributes, frame captions and subtitle. Example:**

**Objects:** First, velouté sauce. Then, béchamel sauce. After that, stock pot. Finally, chocolate milk.
**Events:** First, cooking product. Then, canned food. After that, batter wearing. Finally, cleaning liquid.
**Attributes:** First, greycream. Then, boiled. After that, stirring food. Finally, serving ball.
**Frame Captions:** First, a piece of cake sitting on top of a stove. Then, a blender filled with liquid on top of a stove. After that, a dirty pan sitting on top of a stove. Finally, a man with a bald head with a small animal on it.
**Subtitle:** We're going to start adding in some chicken stock, so just add that straight into the pan.
**Video Caption:** add in some chicken stock in the pan and whisk

**Objects:** First, egg decorating. Then, egg slicer. After that, baking mold. Finally, hollandaise sauce.
**Events:** First, measuring cups. Then, scambled eggs. After that, baking cups. Finally, spilled egg yolk.
**Attributes:** First, spreading butter. Then, purple fondant. After that, yellow frosting. Finally, made of fondant.
**Frame Captions:** First, a person is mixing a mixture in a bowl. Then, a person using a spoon to mix ingredients in a bowl. After that, a blue tray filled with muffins on top of a counter. Finally, a close up of a person making a muffin.
**Subtitle:** And cover the hot dog with the rest of the batter. Bake these.
**Video Caption:** add batter to cover the hot dog

**Objects:** First, cabbage soup diet. Then, moqueca. After that, west african cuisine. Finally, minestrone.
**Events:** First, cooked food. Then, stewed vegetables. After that, meal cooked. Finally, cooking product.
**Attributes:** First, simmering. Then, stirring food.
**Frame Captions:** First, a pot of food sitting on top of a stove.
**Subtitle:** Recipe going to put in about a tablespoon of some parsley. You can add a little bit of olive oil at the end. I like the taste of that myself, and I'm going to put some salt in now. Put any salt in yet and I like to wait till the very end because it does make the. Being a little tough, but in about a table.
**Video Caption:** add parsley olive oil and salt to the pan

**Objects:** First, tabbouleh. Then, ful medames. After that, fattoush. Finally, israeli salad.
**Events:** First, veggie topping. Then, cilantro is green. After that, eaten salad. Finally, people are eating.
**Attributes:** First, mixed into salad. Then, containing salad. After that, removing food. Finally, mixing food.
**Frame Captions:** First, a person is mixing a salad in a bowl. Then, a person pouring a glass of water into a bowl of food. Finally, a close up of a person mixing a salad in a bowl.
**Subtitle:** To add the future bread. Just put it on top. Read it. Like today. By the way, this is a salad that can be combined with maybe 5 Dishel Thought.
Video Caption: add pita bread and mix in

**Objects:** First, mexican food. Then, korean taco. After that, sandwich wrap. Finally, piadina.
**Events:** First, food is served. Then, hand holding food. After that, quesadilla being cut. Finally, some wrapped food.
**Attributes:** First, taco shells. Then, preparing food. After that, wrap ad. Finally, wrap.
**Frame Captions:** First, a woman standing in a kitchen preparing food. Then, a person is putting toppings on a tortilla. After that, a person is putting a tortilla wrap on a plate. Finally, a close up of a person preparing food on a plate.
**Subtitle:** Easy and then just roll it up. I need to get my Gwacham.
Video Caption: roll the burrito up

**Objects:** <query objects>
**Events:** <query events>
**Attributes:** <query attributes>
**Frame Captions:** <query frame captions>
**Subtitle:** <ASR transcript>
**Video Caption:**

Figure 4: An example of few-shot prompts with ASR input for YouCook2 captioning.

**Answer the question based on the objects, events, attributes and frame captions. Example:**

**Objects:** First, jheri curl. Then, orgasm. After that, making out. Finally, special effects.
**Events:** First, man kissing. Then, kiss the frog. After that, lips are together. Finally, couple embracing.
**Attributes:** First, bitten off. Then, pursing lips. After that, one of a couple. Finally, 70s.
**Frame Captions:** First, a man kissing a woman on the cheek. Then, a woman talking on a cell phone in a room.
Question: what are kissing each other?
**Answer:** couple

**Objects:** First, boiled egg. Then, pickled egg. After that, egg slicer. Finally, egg shaker.
**Events:** First, cutting pizza. Then, chopped onions. After that, prepairing food. Finally, scambled eggs.
**Attributes:** First, soft boiled. Then, egg shaped. Finally, cutting food.
**Frame Captions:** First, an egg being sliced on a cutting board with a knife. Then, a person cutting an egg on a cutting board. After that, a person peeling an egg on a cutting board. Finally, a person is peeling an egg on a cutting board.
Question: what is a woman chopping?
**Answer:** egg

**Objects:** First, roar. Then, lion. After that, brazilian terrier. Finally, akbash dog.
**Events:** First, child is playing. Then, moving her tail. After that, running dog. Finally, dog running.
**Attributes:** First, playing catch. Then, cub. After that, eager to play. Finally, jumping up.
**Frame Captions:** First, a baby lion playing with a toy in the grass. Then, a small lion cub playing with a ball. Finally, a baby lion standing on top of a lush green field.
Question: what does a lion try to climb over?
**Answer:** wall

**Objects:** First, vibraslap. Then, indian musical instruments. After that, sound engineer. Finally, saxophonist.
**Events:** First, grooves for racks. Then, man performing. After that, man is playing. Finally, pipes connected.
**Attributes:** First, blue sunny. Then, hifi. After that, playing music. Finally, sax.
**Frame Captions:** First, a man playing a saxophone in a living room. Then, a man playing a musical instrument in a living room.
**Question:** what did the man play the sax in?
**Answer:** room

**Objects:** First, shoe care. Then, hairstyling product. After that, waxing kit. Finally, printer maintenance kit.
**Events:** First, wearing shin guards. Then, guy dipping. After that, man is preparing. Finally, man prepares food.
**Attributes:** First, recycling symbol. Then, dry in background. After that, cleaning supplies. Finally, odor diffuser.
**Frame Captions:** First, a man sitting at a table with a lot of bottles on it. Then, a man sitting at a table with a bunch of bottles on it.
**Question:** what is the man doing?
**Answer:** use

**Objects:** <query objects>
**Events:** <query events>
**Attributes:** <query attributes>
**Frame Captions:** <query frame captions>
**Question:** <query question>
**Answer:**

Figure 5: An example of few-shot prompts for MSVD question answering.

**Predict what is more likely to happen next based on the frame captions and dialogue. Example:**

**Frame Captions:** First, a man in a black shirt is serving food. Then, a group of people in a kitchen preparing food. After that, a man standing in front of a large pan filled with food. Finally, a frying pan filled with corn and a spoon.
**Dialogue:** I took a portion of it, you can smell the onions. You can smell kind of a spice like cinnamon a little bit. He's adding some oil, he's adding in some more onions. Oh, man. speaking in foreign language. - [Mark] Ah, wow, it smells so good. And he's actually gonna make it into a sandwich.
**Question: What is more likely to happen next? A:**He puts the hot sauce on the food. **B:**The chef will serve the food.
**Answer:** The chef will serve the food.

**Frame Captions:** First, a close up of a person wearing a suit and tie. Then, a close up of a person with long hair. Finally, a man sitting at a table with a chess board in front of him.
**Dialogue:** Beckett : Central Park, Washington Square Park. Beckett : Those would be great places to meet up with someone. Beckett : without drawing attention. Castle : Exactly. Now what if each piece stood for the first letter of a word? Bishop for "B." Pawn for "P"? Okay, "B" and then seven spaces. That could be Brooklyn. And Blakely made his phone call from Brooklyn. So, Brooklyn, B-B-P, Brooklyn Bridge Park? That meeting is at 5 : 00. That's in half an hour. Castle : If Blakely shows, we can find out what Pandora is and we can find Gage. Castle : What do you say? Beckett : Blakely should have been here by now. Beckett : Maybe he knows that Tracy's dead. Beckett : or maybe Gage already killed him. Castle : Choose the audacity of hope. I say he'll be here. Then shouldn't you call Sophia? Castle : And look like an ss if I'm wrong? Castle : You know, I have to admit, Beckett : I'm actually kind of surprised that you've never mentioned her before.
**Question: What is more likely to happen next? A:**Beckett plays a move in the game. **B:**Beckett pulls out a knife out of his shoe
**Answer:** Beckett plays a move in the game.

**Frame Captions:** First, a man standing at a podium in front of a class. Then, a woman sitting in a chair in front of a group of people. After that, a woman with long blonde hair sitting in front of a man. Finally, a man standing at a podium in front of a monitor.
**Dialogue:** Ted : Thank you! Ted 2030 : Now... Professor Mosby had arrived. Of course, if I had taken that girl's question... who, by the way,was not your mom. your mom was sitting... Wait, let me finish this story real quick. Here's what that girl would have said. Blond girl : I'm sorry to bother you, Professor Mosby, Blond girl : but this isn't Architecture 101. Blond girl : This is Economics 305. Blond girl : You're in the wrong classroom. Yes, I was in the wrong classroom. And thus began. the most humiliating seven minutes of my life. Ted : Here's your think-about-it for the day. Ted : Every single person in this room. Ted : is already an architect. A girl : Architect?
**Question: What is more likely to happen next? A:**Marshall reads a letter that brings him to tears. **B:**Marshall reads the note to Lily.
**Answer:** Marshall reads the note to Lily.

**Frame Captions:** First, two men standing in a kitchen preparing food. Then, a couple of men standing next to each other in a kitchen. After that, two men standing in front of a large pan of food. Finally, a large pot of food on a table.
**Dialogue:** The ultimate mutton karahi oh look at that.
**Question: What is more likely to happen next? A:**The hosts tell the viewers how good the lobster is **B:**The host tells the camera "I'm ready to try it out".
**Answer:** The host tells the camera "I'm ready to try it out".

**Frame Captions:** First, a man standing next to a desk in a room. Then, a woman in a red jacket talking to a man. Finally, a woman talking to a man in a room.
**Dialogue:** Stuart : - Hey, Leonard. - Oh, hi. - How's it going? - Good, good. Leonard : - You? - Fine. - Oh, yeah, hey, can I ask you something? - Sure. Penny : You know your friend Stuart? Sheldon : Yes. Penny : Well, he asked me out again and I said yes. Penny : And then I started thinking maybe I should talk to you first. - About what? - Well, does it bother you? Penny : Me going out with one of your friends?
**Question: What is more likely to happen next? A:**The girl will agree with Leonard and ask good follow up questions. **B:**Leonard says no it doesn't bother him in an awkward way.
**Answer:** Leonard says no it doesn't bother him in an awkward way.

<Omit five examples here>

**Frame Captions:** First, a woman sitting on a couch holding a bouquet of flowers. Then, a woman sitting in a chair talking to a man. After that, a woman in a gold dress sitting on a couch. Finally, a woman sitting on a couch holding a remote control.
**Dialogue:** Phoebe : So you two were married, huh? Phoebe : What happened, you just drift apart? Do you remember our wedding day? Did you know I slept with the best man? Yes, he told me. At least I think that was what he said. It was difficult to understand with his legs wrapped around my head. Mrs. Geller : Here comes the bride. Phoebe : Oh, my God, Monica!
**Question: What is more likely to happen next? A:**People in the room will tell Monica that she is pretty. **B:**Monica will claim this is the best day ever.
**Answer:**

Figure 6: An example of few-shot prompts for video-language event prediction (VLEP) task.

**Temporal-aware Prompt**

**Generate a video caption based on the objects and frame captions. Example:**

**Objects:** First, closed door. Then, door handle. Then, opened door. Finally, room.
**Frame Captions:** First, man standing next to a wooden door. Then, a close view of a door handle. Then, man in a room.
**Video Caption:** A man is standing in front of a closed door, he reaches for the handle and opens it, and then he walks into a room

**Objects:** First, room. Then, opened door. Then, door handle. Finally, closed door.
**Frame Captions:** First, man in a room. Then, a close view of a door handle. Then, man standing next to a wooden door.
**Video Caption:** A man in a room walking towards the opened door, he reaches for the handle and closes it.

**Objects:** First, night sky. Then, sun moving.
**Frame Captions:** First, stars shining at night. Then, sun over an ocean.
**Video Caption: The stars are shining at night, and the sun is rising over the ocean.**

**Static Prompt**

**Generate a video caption based on the objects and frame captions. Example:**

**Objects:** closed door, door handle, opened door, room.
**Frame Captions:** man standing next to a wooden door. a close view of a door handle. man in a room.
**Video Caption:** A man is standing in front of a closed door, he reaches for the handle and opens it, and then he walks into a room

**Objects:** room, opened door, door handle, closed door.
**Frame Captions:** man in a room. a close view of a door handle. man standing next to a wooden door.
**Video Caption:** A man in a room walking towards the opened door, he reaches for the handle and closes it.

**Objects:** night sky, sun moving.
**Frame Captions:** sun over an ocean, stars shining at night.
**Video Caption: The sun is setting over the ocean as the stars start to shine in the night sky.**

Figure 7: Full prompt of the "Sunrise" scenario for the example shown in Figure 3 in the main body. We show the impact of temporal-aware prompt on capturing temporal dynamics in videos. The sentences in blue following "Video Caption:" are generated by GPT-3. Text marked in green indicates the generated caption is semantically coherent with the given objects and frame captions, while text marked in red indicates incorrectness.