# OpenReview forum: "Language Models with Image Descriptors are Strong Few-Shot Video-Language Learners"
_NeurIPS.cc/2022/Conference — NeurIPS 2022 Accept_

### Official Review · Reviewer_An3y · 2022-07-06

**Rating:** 4
**Confidence:** 5
**Soundness:** 2 fair
**Presentation:** 2 fair
**Contribution:** 2 fair

**Summary:**

This paper proposes a few-shot Video-language Learner termed VidIL based on image- language modeling. A temporal order aware prompt is also proposed by decomposing videos into a hierarchical structure. Without pre-training on videos, the proposed VidIL is evaluated on a variety of few-shot video-to-text tasks, such as domain-specific captioning, question answering, and future event prediction.

**Questions:**

See my questions in weaknesses.

**Limitations:**

Yes.

**Strengths And Weaknesses:**

Strengths:
+ A few-shot Video-language Learner termed VidIL is proposed based on image- language modeling.
+ A temporal order aware prompt is proposed by decomposing videos into a hierarchical structure.
+ A variety of few-shot video-to-text tasks are included for performance evaluation.

Weaknesses:
- The technical novelty of this paper is very limited. CLIP, BLIP, and InstructGPT are respectively deployed for visual tokenization, frame captioning, and conditional text generation with few-shot prompt. Excluding this, the contributions made by the authors are rather minor. I even have doubt that the performance of VidIL is mainly due to the deployment of CLIP, BLIP, and InstructGPT.
- The proposed VidIL can still be considered as prompt learning for large-scale cross-modal pre-training models. Therefore, other prompt learning methods should be included in the main experiments. Some examples are:\
[a] Align and Prompt: Video-and-Language Pre-training with Entity Prompts, arXiv:2112.09583, 2021.\
[b] CPT: Colorful Prompt Tuning for Pre-trained Vision-Language Models, arXiv:2109.11797, 2021.
- The method part should be reorganized in a more formal way. Although the source code is given, an outlined algorithm would help the readers easier to capture the proposed method.

---

> ### Author Response · Authors · 2022-08-02
> **Our contribution is on transferring the generalization ability of language models to video language tasks and we have compared with existing prompt baselines**
>
> We thank Reviewer #4 for the detailed comments. We appreciate the reviewer’s critical feedback on the approach.
>
> ***The technical novelty of this paper is very limited. The performance of VidIL is mainly due to the deployment of CLIP, BLIP, and InstructGPT?***
> - As mentioned in the Introduction, our primary goal is to **investigate the possibility of transferring the strong generalization ability from large language models WITHOUT any video data and additional parameter tuning**.
> - We are the first to demonstrate language models’ ability in video understanding. In order to make language models understand videos, we further propose to use image-language foundation models to generate the textual representation of a video.
> - We want to reiterate **the challenges that is unique for videos**: (1) the temporal dimension: we handle it using temporal aware prompt (2) the multiple levels of semantics: we handle it by visual tokens of multiple levels and captions to describe the semantics of the entire scene (3) the speech modality: we handle it by the unified text representations.
> - Finally, the main message here is that even though this language model based framework does not see any videos and without any fine-tuning, it can still achieve very strong few-shot performance on video-language tasks, especially for the tasks requires semantic reasoning, such as future event prediction.
>
> ***Comparison with other prompt learning methods***
> - As shown in Table 3 and Table 5 in the original version, we already included the results of ALPRO (Align and Prompt), but they are not comparable since they require a large number of videos for fine-tuning, as dimmed in gray.
> - CPT (Colorful Prompt Tuning) is working on visual grounding, cannot perform video captioning, question answering and future event prediction. And we want to reiterate that our method requires no parameter tuning.
>
> ***The method part should be reorganized in a more formal way***
> - Equation 1 includes the formula for the language modeling objective. Since there is limited space, we present the algorithm in the form of an illustration in Figure 2, which we believe makes it easier to follow.

---

> > ### Comment · Reviewer_An3y · 2022-08-07
> > **Response to authors' rebuttal**
> >
> > Thanks for the detailed response. Most of my concerns have been addressed.
> >
> > As the authors claimed, the main contribution of this paper is to transfer the generalization ability of language models to video-language tasks. However, this is mainly due to utilizing CLIP to generate the textual representation of a video. My question is: there is any novelty in the deployment of CLIP? Please clarify it.

---

> > > ### Author Response · Authors · 2022-08-07
> > > **The novelty of the deployment of CLIP**
> > >
> > > Thank you for your feedback and for confirming that the questions have been addressed. The novelty of the deployment of CLIP can be summarized as follows:
> > >
> > > ***(1) Additional visual tokens to address the limitations of the expressiveness of the universal textual representations (lines 123-126):***
> > >
> > > The preliminary design of this model is using an image captioner in BLIP and just feeding frame captions into GPT-3. However, one major difficulty of few-shot video understanding is that the annotators have different standards and granularities of describing videos. For a video of many people on the beach, one annotator might say there are many people on vacation, and another one might mention that there is a boy playing in the sands. The motivation that we add CLIP is to generate visual tokens is to provide visual features of different granularities, so as to alleviate the limitation of the expressiveness of textual representations.
> > >
> > >
> > > ***(2) The key challenge of applying CLIP is to achieve a trade-off between visual token quality and coverage. We achieve this by linguistic knowledge:***
> > > - ***To optimize COVERAGE: Multiple levels of granularities, including objects, events and attributes (lines 139-141):***
> > >
> > > There are multiple ways to generate visual tokens, but we found that pre-defined classes for classification, such as those in ImageNet, are far from enough for covering the rich semantics in open-domain videos, so we adpot retrieval based method. We further decompose the visual tokens into nouns (objects), verbs (events) and adjectives (attributes) and compose the vocabulary from these three angles.
> > > - ***To control the QUALITY: Guidance from Semantic Role Labeling (lines 150-155):***
> > >
> > > To control the quality, we leverage Semantic Role Labeling. We first select the phrases that contain at least one verb and one argument as events. Then we remove highly similar events based on their sentence similarity using Sentence BERT embeddings.
> > >
> > > ***(3) The conclusion of vocabulary selection (lines 156-157):***
> > >
> > > We found that using large but noisy vocabulary is more effective than using small but clean vocabulary in our retrieval-based setting with CLIP.
> > >
> > >
> > > ***(4) Highly modular: CLIP can be replaced with other models:***
> > >
> > > Any further advanced model can and is expected to replace CLIP, to boost the performance further. The key idea is to get visual tokens following our granularities and filtering strategy, but not limited to any specific image foundation models.
> > >
> > > ***(5) Specially designed for leveraging language models to connect dots (lines 164-168):***
> > >
> > > The way of extracting visual tokens is based on the power of large language models. As the visual tokens do not make sense on their own, the power of language models allows us to connect the dots. Recent advances in image foundation models and large language models are key to the success of this new paradigm. As a result, we believe this new paradigm will serve as a strong competitor for various video tasks in the near future.
> > >
> > > In summary, the key idea is to produce visual tokens that can achieve **a trade-off between quality and coverage**. In addition, the exploration of **different granularities of text representations**, as well as the **guidance from Semantic Role Labeling**, can serve as a good basis for this new line of work utilizing language models for video analysis.

---

### Official Review · Reviewer_12Dy · 2022-07-10

**Rating:** 6
**Confidence:** 5
**Soundness:** 4 excellent
**Presentation:** 4 excellent
**Contribution:** 3 good

**Summary:**

This paper presents a few-shot learning approach for video and language tasks.
The approach relies on pre-trained and frozen models: first, vision models are used to extract for each video frame: (A) a caption and (B) a visual attributes (Objects, Events, Attributes) from a set of pre-defined ones.
A text-only prompt is then constructed showing few-shot examples of how to perform a vision-language task utilising information the generated captions and attributes.
Finally, the prompt is fed to a large language model (InstructGPT) to solve the task defined by the few-shot examples.
The model is evaluated on several video benchmarks such as: video captioning, video question answering and next event prediction.

**Questions:**

1. About the use of temporal marker: how does it work when you have more frames than temporal markers?
2. I was curious to know the impact of the choosen language models. Why did you choose InstructGPT over GPT-3? Is it because InstructGPT is much more lightweight to run? Or is it because InstructGPT is performing better than GPT-3 at following the few-shot examples?

typo:
L 276 : ‘video-captain’


**Limitations:**

The authors have adequately addressed the limitations and potential negative societal impact of their work

**Strengths And Weaknesses:**

**Strength**:

- The approach is relatively easy to follow
- One strength of the approach is that it does not rely on video data for training. More precisely, it does not require any training at all (in the sense of gradient descent training) as the approach mostly builds on top of pre-existing pre-trained/foundational models.
- The VLEP (Next event prediction task) results are strong and they provide a new state-of-the-art number on that task.
- The method is more interpretable than standard VLM models.
- The approach is highly modular. It is straightforward to update it when improved captioning models or language models are available.

**Weaknesses**:

- I believe the approach to be highly bottlenecked by the expressiveness of the vision foundational models. As a concrete example, I do not see how this approach can work well on traditional action recognition benchmarks (i.e. Kinetics or Something Something) when the task requires low-level temporal understanding of actions. Also the approach does not work for task where there is no useful signals to leverage from frame captions and visual attributes. For instance such task could be fine-grained spatial visual question answering. MSVD-QA and MSRVTT-QA are not really fine-grained VQA dataset, so there are no questions requiring spatial understanding such as "'What is on the left of the car?' or 'What is on bottom the table?'. Right now I do not see how the provided frame captions and attributes are enough to answer fine-grained question about the visual content.

- Why are the ablation studies only reported on MSVD-QA? The results would have been more convincing with several evaluation datasets.
For example in Table 7, we do not see clear improvement when the number of shots increase. As the ablation study is only reported on one dataset, I'd assume the conclusion were not even better on the other datasets? Also I find that the number of in-context examples VS the true few-shot number in this Table 7 is not clear. It would have been clearer to have two columns, one for both. It also look like the in-context selection is not really beneficial for more than 5 shots.

- Table 3: Clarify the real number of shot used in that table. What is the number of in-context vs the number of shots the retrieval system has access to. Is it 5-shot in the sense of 5 in-context shot?

- Table 6: It looks like frame captions are enough to get most of the useful signal, as it only improves the MSVD-QA top-1 accuracy of 1.2% when adding the different visual attributes. This is disappointing knowing all the emphasise around the need for the visual attributes in the approach section.

- In the Related work section, the authors wrote L 110:
‘We are the first work to leverage prompting a frozen language model for tackling few-shot video-language tasks with a unified textual representation’.  I believe this is an overclaim as [1] have already done something in a similar spirit. It looks to me this paper, mostly extend the work of [1] with non-vqa tasks and video instead of images.

[1] Yang, Z., Gan, Z., Wang, J., Hu, X., Lu, Y., Liu, Z., & Wang, L. (2022, June). An empirical study of gpt-3 for few-shot knowledge-based vqa. In Proceedings of the AAAI Conference on Artificial Intelligence (Vol. 36, No. 3, pp. 3081-3089).

---

> ### Author Response · Authors · 2022-08-02
> **We focus on the unique challenges of video understanding and the benefit of transferring language models' generalization abilities outweighs the limitations**
>
> We thank Reviewer #3 for the detailed and insightful comments. We appreciate the reviewer’s encouraging remarks on the strengths of the approach.
>
> ***Bottlenecked by the expressiveness of the vision foundational models***
>
> - This is true. We agree with it and we are aware of the well-known trade-off for using a universal textual representation on vision modality.
> - However, the benefit of transferring language models' generalization abilities generally outweighs their limitations.
>     - The main reason we choose to adopt the image-language foundation models is that we can transfer the strong generalization ability from large language models to videos WITHOUT any additional training.
>     - The limitation is obvious that we might lose some low-level vision features which can be crucial for fine-grained tasks.
>     - Alternatively, another line of work such as Flamingo focuses on building end-to-end visual language models, but it will require intensive pre-training on a large amount of vision-language data.
> - In this work, we aim to show that we can actually make language models perform video tasks without any additional parameter tuning and can already get very strong performance.
>
> ***Comparison with related work***
>
> - We are the first to focus on videos compared with the above related work, since video processing has two main challenges to address compared to images:
>    - (1) the temporal dimension, we handle it using temporal aware prompt;
>    - (2) the multiple levels of semantics, we handle it by visual tokens of multiple levels and captions to describe the semantics of the entire scene;
>    - (3) the speech modality, we handle it via the unified text representaions.
>
> ***Ablation studies only reported on MSVD_QA***
>
> - We only perform ablation studies on MSVD_QA mainly because of the inaccessible cost and time for exhaustively running  large-scale experiments all benchmarks involving GPT-3.
> - Taking a step back, we select MSVD_QA as our ablation benchmark for two reasons: (1) it is relatively small in scale; (2) video question answering is a representative video-to-text task since other tasks can easily be reformulated into a QA task, for example video captioning.
>
> ***In-context Selection Details***
>
> - We have updated Table 6 in the revision version to make it more clear.
> - In the setting without in-context selection, the number of shots equals to the number of in-context examples that will appear in the prompt.
> - In this setting with in-context selection, we choose 5 in-context examples from the number of shots (which can be 10, 20, etc).
> - That’s why the performance of using in-context selection could be lower than using all of the 10-shots or 20-shots. However, we want to emphasize that the in-context selection also improves efficiency.
> - Furthermore, the fact that 30-shot w/ in-context selection outperformed using all 30-shots shows that in-context selection can help choosing better examples when the set of examples might be somewhat noisy.
>
> ***Frame captions are enough to get most of the useful signal***
> - This is indeed the case for most of the current video-language benchmarks. This phenomenon is actually raised by previous work such as ClipBert [1].
> - However, as shown in the example of Figure 2, there are some cases where salient information only appears in the visual tokens (e.g., bandage). Since these cases are not dominant in the benchmarks, the scores only show mild improvements.
>
> ***Details of temporal markers***
> - We dynamically apply temporal markers "First,", "Then,", "After that,", and "Finally," to different numbers of resulting frames.
> - If we have more than 4 frames, we repeat "Then" or "After that" for frames that are not the first or the last.
> - If we have less than 4 frames, we remove “Finally”, “After that”, etc.
>
> ***InstructGPT over GPT-3***
>
> - InstructGPT shows better performance than GPT-3 on following instructions, which is essential for our few-shot tasks.
>
> ***References***
>
> [1] Lei, Jie, et al. "Less is more: Clipbert for video-and-language learning via sparse sampling." Proceedings of the IEEE/CVF Conference on Computer Vision and Pattern Recognition. 2021.

---

> > ### Comment · Reviewer_12Dy · 2022-08-07
> > **Reply to author rebuttal**
> >
> > I would like to thank the authors for the clarification brought into the rebuttal.
> > I am happy to keep my overall rating to Weak Accept.
> > I believe that, although one can see this approach as an engineering pipeline connecting together different pretrained / foundational models via a language interface, it a valuable baseline for few-shot video and language tasks to consider due to its highly modular and conceptual simplicity of its design. I especially appreciated the fact no video training was used.

---

> > > ### Author Response · Authors · 2022-08-08
> > > **Thank you for the feedback**
> > >
> > > Thank you for the detailed feedback and confirming the questions have been addressed. We appreciate your highlights on the strength of the proposed framework, including does not rely on video data for training, strong results and new state-of-the-art for future event prediction, better explainability and being highly modular.

---

### Official Review · Reviewer_6Be3 · 2022-07-11

**Rating:** 3
**Confidence:** 5
**Soundness:** 1 poor
**Presentation:** 2 fair
**Contribution:** 1 poor

**Summary:**

This paper proposed a new model for video-language tasks with few-shot ability. The task is decomposed into hierarchical level, to first translate objects, frame captions and their relations using CLIP model. Then a langauge model is instructed with prompts to compose the video frame.

**Questions:**

1- Line 28: Only the encoder model with bidirectional attention are limited to understanding tasks, but with causal attention, they are capable of both understanding and generation. In fact, autoregressive models have shown promising few-shot performances, because they are more parameter efficient in compare to encoder-decoder models

2- Is the few-shot ability arises from using language model at the end, or from decomposing the tasks to two models?

3- Line 116: it is mentioned that the proposed model is not required to be pretrained on any video, compare to Flamingo. However, it is mentioned otherwise in other parts of the paper!

4- is there an ablation on the decomposition of video contents into frame, visual token level, video level on the few-shot performance?

5- Figure 2: in visual token level, each frame includes multiple objects. How the template format organizes objects of one frame in order. how the order of objects is defined for each frame? (refer to “few-shot context” box). Is there an ablation on the ordering of visual tokens?

6- is there an ablation on different task instruction formats? how does wording of the template affects performance?

7- Table 7: increasing shots also increase standard deviation. why model performance variance increase by more training examples?

**Limitations:**

The proposed model is not end-to-end. It is based on using CLIP and InstructGPT3. Therefore, there is minimal novelty in the architecture.

**Strengths And Weaknesses:**

The proposed model is not end-to-end. It is based on using CLIP and InstructGPT3. The approach is simply based on extracting candidate objects and captions frame by frame using CLIP encoder, and then creating a prompt to feed to InstructGPT3 using in-context learning to achieve few-shot performance. Therefore, there is minimal novelty in the architecture

---

> ### Author Response · Authors · 2022-08-02
> **The primary goal of this paper is not to propose an end-to-end model that can perform large-scale pretraining or finetuning, but to transfer the strong generalization ability of language models to video understanding**
>
> We thank the reviewer for the detailed comments. We appreciate the reviewer’s critical feedback on the approach.
>
> ***Proposed model is not end-to-end***
>
> - We appreciate your review and detailed suggestions, but we respectfully disagree with you on this point.
> - We would like to reiterate that the main objective of this paper is not to build a large end-to-end model that requires extensive pre-training or fine-tuning on a large amount of vision-text data, and **we argue that it is unnecessary to make the end-to-end constraint given the focus of efficiency and performance**.
> - As mentioned in Introduction, **our primary goal is to investigate the possibility of transferring the strong generalization ability from large language models WITHOUT any video data and additional parameter tuning**.
> - In order to make language models understand videos, we further address the challenges that is unique for videos:
>     - (1) the temporal dimension, we handle it using temporal aware prompt;
>     - (2) the multiple levels of semantics, we handle it by visual tokens of multiple levels and captions to describe the semantics of the entire scene.
> - The main message here is that even though this framework does not see any videos and does not need any fine-tuning, it can still achieve very strong few-shot performance on video-language tasks, even outperform large-scale video pretraining models. It is a new way to process and understand videos.
>
>
> ***Autoregressive models have shown promising few-shot performances***
>
> - As mentioned in line 30-32, the major limitation of autoregressive models with casual attention is that they still require a large number of task-specific video-text pairs to finetune.
> - Experiments of VinVL also demonstrate it highly relies on  finetuning and does not perform well when transferring to different benchmarks.
> - In contrast, our model requires ZERO fine-tuning and achieves strong performance on various benchmarks.
>
> ***Few-shot ability arises from using language model or decomposing tasks***
>
> - Following general few-shot definition, the goal is to be able to perform the task with a few task-specific examples, so the few-shot ability of various tasks is from language models, i.e., captioning, question answering, future event prediction, etc. The fact that VidIL outperforms single BLIP as shown in Table 1 and Table 2 verifies this point.
> - Furthermore, there is another few-shot (zero-shot) ability in our paper, namely processing videos without training on any videos, and this ability is largely dependent on the interaction with image-language models such as BLIP and CLIP. It can be viewed as CLIP/BLIP enables language models to transfer its few-shot ability to video tasks.
>
> ***Ablation on the decomposition of video contents into frames, visual token level***
>
> - We already included this ablation in Table 6 in the original version (Table 5 in the revision version), where adding visual tokens generally improves the performance.
>
> ***Ordering of the visual tokens***
> - We decide the visual token order based on the position of each frame from which it is extracted. If a visual token is extracted in more than two frames, we use the average of the position index of those frames.
>
> ***Wording of the task instruction***
> - Our task instruction prompt is largely following [1] which has shown to be effective in incorporating prompted knowledge.
> - We did not try optimizing the wording of the prompt due to the high cost and difficulty of tuning templates for InstructGPT.
> - It is also not our main focus to optimize the wording of prompt templates. There is a surge of studies dedicated to finding good prompt templates, but our primary focus on the prompt design is to investigate how to represent a video into textual prompts covering multiple levels of video semantics.
> - As a result, we aim to keep the templates with the most concise and straightforward wording, and exclude complicated designs.
> - Figure 3 illustrates a qualitative analysis of prompt design in which we found that instead of the wording, the temporal orders has an impact on the final output.
>
>
>
>
>
> ***Table 7: why model performance variance increase by more training examples?***
>
> - The high variance across different selection and permutation of the in-context examples has been demonstrated by various previous work such as “Calibrate Before Use” [2].
> - The variance persists with more data and larger models. This is exactly the reason why we further adopt the in-context selection which we found to be effective in reducing the variance and can even result in better performance, as shown in Table 7 (Table 6 in the revision version).
>
> ***Reference***
>
> [1] Liu, Jiacheng, et al. "Generated knowledge prompting for commonsense reasoning." arXiv preprint arXiv:2110.08387 (2021).
>
> [2] Zhao, Zihao, et al. "Calibrate before use: Improving few-shot performance of language models." International Conference on Machine Learning. PMLR, 2021.

---

> > ### Comment · Reviewer_6Be3 · 2022-08-08
> > **Answer to rebuttal**
> >
> > Thanks for addressing the question.
> >
> > Overall, I think the proposed architecture shows that combination of clip and instructgpt3 can be used as a baseline for zero-shot video understanding. But this combination has minimal technical novelty to be accepted as a new model for this task.
> >
> > the framework is using the generalization ability of CLIP model for Image understanding, and language generalization of InstructGPT3 for prompt generation. There is no ablation on how much each model contributes to generalization.
> >
> > Since the model is using InstructGPTT3, it is costly to ablate on the prompt format (as author mentioned in rebuttal), which indicates another downside of the pipeline approach by combining models.

---

> > > ### Author Response · Authors · 2022-08-08
> > > **Main novelty & Contribution of individual components**
> > >
> > > Thank you for your feedback and for confirming that the questions have been addressed. We address the further comments as follows:
> > >
> > > ***Main novelty***
> > >
> > > - As you mentioned, the main contribution of this work is to leverage the generalization ability from language models to achieve strong few-shot performance on video-language tasks without any video data or parameter tuning. **We would like reiterate that the combination of the three foundation models to achieve good performance is not trivial**, and simple combination will suffer from the expressive ability of text representations or include too much noise to perform well.
> > > - The key idea is to produce unified text representations that can **achieve a trade-off between quality and coverage**, to provide enough and relatively precise information for language models. The exploration of **different granularities of text representations**, as well as the **guidance from Semantic Role Labeling**, can serve as a **good basis for this new line of work utilizing language models for video analysis**.
> > > - Video processing faces unique challenges including: (1) temporal dimension (2) multiple levels of semantics (3) additional modalities including speech.
> > > - We propose dedicated solutions for these unique challenges including:
> > >     - (1) **a hierarchical textual video representation (Section  3)**:
> > > One major difficulty of few-shot video understanding is that the annotators have different standards and granularities of describing videos. For a video of many people on the beach, one annotator might say there are many people on vacation, and another one might mention that there is a boy playing in the sands. **There are multiple ways to convert videos to text representations, and our hierarchical design is achieving a trade-off between text quality and coverage.** We divide visual features into different granularities and borrow linguistic knowledge, so as to alleviate the limitation of the expressiveness of textual representations.
> > >     - (2) **structure-aware visual tokenization with guidance from Semantic Role Labeling (Section 3.2)**:
> > > To provide visual tokens of high-quality and high-coverage, we leverage Semantic Role Labeling. We first select the phrases that contain at least one verb and one argument as events. Then we remove highly similar events based on their sentence similarity using Sentence BERT embeddings. To optimize coverage, we propose visual tokens of multiple levels of granularities by decomposing the visual tokens into nouns (objects), verbs (events) and adjectives (attributes). Also, we found that using large but noisy vocabulary is more effective than using small but clean vocabulary in our retrieval-based setting with CLIP.
> > >     - (3) **temporal-aware prompt (Section 3.3):**
> > > The role of language models in this framework is connecting the dots (noisy visual tokens and sentences), and get a coherent story following temporal orders, or perform reasoning tasks such as future event prediction. Actually visual tokens do not make sense on their own, the power of language models perfectly fill in here. Recent advances in image foundation models and large language models are key to the success of this new paradigm. As a result, we believe this new paradigm of using unified text representations will serve as a strong competitor for various video tasks in the near future.
> > >
> > > ***Contribution of individual foundation models***
> > >
> > > - In Table 2, 3 (in the original version), we have already compared the performance of using only an image-language model (BLIP) and our full model (which contains BLIP as a component), which showed the significance of adding GPT-3 to connect the dots.
> > > - In Table 6 (in the original version), we further showed that adding visual tokens from CLIP generally improves the performance and reduces the variance.
> > > - In Table 5 (in the revision version), we added a new ablation regarding the performance using only one frame, showing the significance of the temporal dimension.
> > > - Also we want to emphasize that without our framework, single CLIP or single GPT-3 in its nature can not perform any video-language tasks. Visual tokens do not make sense on their own, the power of language models perfectly fill in here to connect the dots and get a coherent story, or reason about future events.
> > >
> > > ***High cost of ablating prompt format***
> > >
> > > - We agree that ablating prompt template for large-scale language models such as GPT-3 is costly. And this problem is widely applied to any model that include large-scale LM as one of its components.
> > > - One good future direction is to investigate efficient language models and semi-parametric models to alleviate this problem.
> > > - In this work, we try to focus on ablating the novel designs, e.g., visual tokens, of our model instead of the wording of the templates. However, we agree that how to efficiently find the best prompt template in this setting is an important research problem and deserves dedicated attention.

---

### Official Review · Reviewer_Y7bx · 2022-07-11

**Rating:** 6
**Confidence:** 4
**Soundness:** 3 good
**Presentation:** 3 good
**Contribution:** 2 fair

**Summary:**

This paper introduces a method for few-shot learning for video understanding tasks. In a nutshell, it works by first extracting information from a given video using pre-trained image and language models: it uses CLIP to extract "visual tokens" for objects, events and attributes, as well as a captioning model (BLIP) to extract image description of a few frames of the video. These automatically generated annotations are then assembled together to form a textual description representing the video by notably keeping some information about the order of events. Since videos are represented by simple text, it allows to use the few-shot capability from an existing language model (InstructGPT in this case) to do few-shot learning on video understanding tasks: video captioning and video question answering. The authors evaluate this approach on five different benchmarks. The model is also used as a pseudo labeler to improve video-text retrieval system.


**Questions:**

**Clarifications**

- L144: "We then select top 5 visual tokens": is it top 5 per frame, or top 5 total? In the Figure 2, it seems there are only 3 for the first frame. Can the authors clarify this? It is also said that 8 frames are used but when I check the Appendix, I can see that there are at most 4 temporal description (First, Then, After that, Finally). Is there some filtering happening on top of the top 5?

- L224: For the BLIP baseline, is there additional training involved or is the proposed technique of "stitching the sampled frames" is a pure inference only trick? This was not clear to me as the term "loss" was mentioned. Can the authors clarify what is meant here?

- Table 2: As far as I know UniVL and BLIP are not "few-shot" methods. How do the authors obtain 10-shot results for these baselines?

- Table 2: How come VidIL can be less good than BLIP (for MSRVTT) knowing that the BLIP captions are used as input in the prompt? Is that because the language model is not able to properly use the information provided by the BLIP model in that case?

- When using in-context selection, are you also ordering the samples in a certain way to account for the recency bias in big language models? (putting the most similar last in the prompt might help)

**Additional ablations**

- How important is the "temporal aware prompt" technique quantitatively? Unless I missed that ablation I could only see the qualitative example of Figure 3. Since this trick seems one of the contribution of the paper, I believe it would be important to properly demonstrate that it brings something.

- How does the performance change with the number of sampled frames?

- How does the performance evolve with the number of shots? What is the maximum number of shots you can fit in the usual prompt of the model?

**Questions about experiments**

- In Table 6, adding event seems to hurt performance (Line 3). Can you provide the numbers for "Frame+Object+Attribute"?
- Table 7: is that possible to put a column "Prompt size" and a column "Number of shots" to better distinguish between these two parameters. I think it will make it clearer. How come in-context selection can actually hurt?


**Limitations:**

Some important limitations are not mentioned:

- Performance potentially highly bottlenecked by the image level "text" extraction
- Replacing videos by text in the first place is limited and may not scale well when moving to longer videos (the approach would always be limited by the size of the prompt of the language model). Having Large Language models that could directly attend high level features instead could help alleviate this issue.

The authors do discuss the societal risk, notably the ones inherited from large language models that might be amplified here by using them on videos.

**Strengths And Weaknesses:**

**Strengths**

- The paper is clearly written and easy to follow
- Few-shot learning in multimodal setting is a very important endeavor
- The proposed method is a good example of repurposing a large language model to do few-shot learning for videos in a simple way that does not involve any training with video and language dataset. It provides a very strong baseline for such models.


**Weaknesses**

- A few clarifications are required around the generation of the prompt (see Question section)
- Conceptually, the approach is quite similar to the PICA paper [61]. It would be good to better clarify the difference with that approach? Is it mainly the video setting and the fact that one has to deal with temporal information? If yes, then would it be possible to show what happens if only a single frame from each video is used instead of multiple ones?
- Some ablations are missing (see Question section for specific suggestions)
- Some results seems a little bit inconsistent and raise some questions (see Question section for more details)
- Should Flamingo be also compared against in Table 2 for consistency (since they provide results for YC2 and VATEX as well?)
 - It would be interesting to see how the performance evolve with number of prompts for all benchmarks but mainly 5 shot results are given (see request in the Question section)
- Limitations of the approach:
  - This approach is brittle as it is bottlenecked by the extraction of visual entities and the captioning model.
  - Such approach may not scale well to long term video understanding (mentioned in the conclusion) since one would be quickly limited by the size of the prompt of Language Models. Also temporal information is only given with words like "First", "After"... How would that be scale to longer videos? Could the authors comment on that?


**Summary of review**

Overall the paper introduces an interesting method and has good results on video benchmarks. However there are a few important questions that are important to address in the rebuttal to help confirm my current weak accept decision.

---

> ### Author Response · Authors · 2022-08-02
> **Our framework focuses on the unique challenges of video understanding**
>
> We thank Reviewer #1 for the constructive and highly detailed comments. We appreciate the reviewer’s positive feedback on the main novelty of our model design.
>
> ***Difference with PICA***
>
> The key differences from PICA are two fold:
> - First, we focus on videos, as well as attaining flexibility in incorporating additional modalities such as ASR. **Video processing faces three main challenges**:
>     - (1) temporal dimension: we handle it with temporal aware prompts;
>     - (2) multiple levels of semantics: we use visual tokens of multiple levels and captions to describe the entire scene's semantics;
>     - (3) multiple modalities including speech: our unified text representation enables it to easily add speech modality.
> - Furthermore, PICA only focuses on VQA while we consider a wide range of video-to-text generation tasks, such as VLEP, where we beat the fully-supervised SOTA with only 10-shots.
>
> ***Comparison with Flamingo in captioning tasks***
>
> - We updated Table 1 (in revision version) including Flamingo results.
> - Compared to Flamingo, we would like to emphasize two observations:
>     - (1) With ZERO video data and ZERO parameter tuning, we can already achieve competitive performance;
>     - (2) Our framework can easily utilize extra modalities such as ASR to obtain significantly better few-shot performance then Flamingo on certain tasks such as Youcook2.
>
> ***Bottleneck of using image-level textual representation***
>
> - We agree and are aware of the well-known trade-off for using a universal textual representation on vision modality.
> - However, we believe the benefit of transferring the strong generalization ability from large language models outweighs the limitation.
>     - As mentioned at line 37 in the original version, the major benefit of this approach is that with the help of recent advances in image-language foundation models, we can transfer the strong generalization ability from large language models to videos without any additional training.
>     - The limitation is that we might lose some low level vision features.
>     - Alternatively, another line of work such as Flamingo [1] focuses on building end-to-end visual language models, but it requires extensive pre-training. In this work, we aim to show that we can actually make language models perform video tasks without any additional parameter tuning and can already get very strong performance.
> - We have added more discussion about it in the limitation section.
>
> ***Scale to longer videos***
> - One possible solution is to solve this in a hierarchical manner.
> - We can first segment a long video into salient clips, individually represent and prompt each clip, and then use a “meta-prompt” to aggregate the information from different clips. Since the textual representation is highly flexible, this process can be easily extended from our current design.

---

> > ### Author Response · Authors · 2022-08-02
> > **Details of experiments and more ablation studies are added**
> >
> > ***Details of prompt generation***
> > - We first select the top 5 visual tokens per frame, and then aggregate them across 8 frames for final video-level visual tokens.
> > - The number of final visual tokens per video is empirically set at 4. However, this number is flexible since we use a dynamic temporal prompt, where additional middle frames can be added by repeating the temporal marker "then". We added this detail into the Appendix.
> > - Due to space limitations, some tokens are omitted in Figure 1 and Figure 2. We have updated the figures in the revision with clear ellipsis whenever there is an omission.
> >
> > ***Few-shot training of BLIP and UniVL baselines***
> > - We use the same randomly selected videos that are used in our few-shot prompt to further fine-tune BLIP and UniVL baselines.
> > - The training details are presented in Appendix C in the original version.
> >
> > ***BLIP perform better in BLEU_4 and CIDEr on MSRVTT***
> > - As mentioned in Section 4.1, VidIL uses BLIP captioning checkpoint (BLIP_cap in Table 1) instead of the original “BLIP” since preliminary experiments show that BLIP_cap results in better frame captions.
> > - VidIL consistently outperforms BLIP_cap which shows that GPT-3 can actually utilize the lower-level information pretty well.
> > - We believe the reason why BLIP can be particularly good at MSRVTT is because of the bias in the caption distribution. As we can see, BLIP outperforms BLIP_cap on MSRVTT but not on VATEX, which shows that different datasets have their certain bias on the “style” of the caption. That’s why we show the average CIDEr [2] across all datasets to mitigate this bias and more faithfully reflect the power of a model.
> >
> > ***In-context example ordering***
> > - Yes, we reorder the selected examples in ascending order based on the similarity score to account for the recency bias [3] in large language models.
> >
> > ***Impact of the number of shots for all benchmarks***
> > - We show the ablation studies with the number of shots only on MSVD_QA due to the inaccessible cost and time for running exhaustive experiments on all datasets using GPT-3.
> > - We select MSVD_QA as our ablation benchmark for two reasons: (1) it is relatively small; (2) video question answering is a representative video-to-text task since other tasks, such as video captioning, can be easily reformulated as a QA task.
> > - The performance generally increases with the number of shots in Table 6, but the benefit starts to saturate at around 30 shots.
> > - Furthermore, the maximum number of shots that can fit into the prompt is highly dependent on the task, for example VLEP task tends to have long dialogue, thus the maximum number of shots is  less then other tasks such as VQA.
> > - We have added these discussions in the revision.
> >
> > ***Adding ablation on Frame+Object+Attribute***
> > - We have added this results in Table 5. Compared with adding all types of visual tokens, the accuracy is slightly better but the variance is worse.
> >
> > ***Ablations on temporal aware prompt***
> > - Temporal dynamics is important to videos and cannot simply represent them as one static frame.
> > - We have added an ablation study in Table 5 showing the results regarding single frame. There is a clear drop compared with using 4 frames (our default setting), indicating the model’s ability to incorporate information from multiple timestamps.
> > - We have also added an ablation study to reverse the ordering of frames in Table 5. The performance only decreased marginally since most current video-language benchmarks, including MSVD, rarely require fine-grained temporal modeling to perform well, and may not be sufficient in reflecting benefits from better temporal ordering. However, we see significant differences in the description coherence during qualitative analysis, where Figure 3 is an example. As mentioned in the future work, we intend to extend this idea to script learning and storytelling where the temporal ordering will be more significant.
> >
> > ***Prompt size vs Number of shots***
> > - We have updated Table 6 to make it clear. We also want to emphasize that the in-context selection increases efficiency. In-context selection uses only five examples to select shots from 5 to 30 (line 210-219 in the original version). That’s why the performance could be lower than using all of the 10-shots or 20-shots. The fact that 30-shot w/ in-context selection outperformed using all 30-shots shows that in-context selection can help choosing better examples when the set of examples are noisy.
> >
> > ***Reference***
> >
> > [1] Alayrac, Jean-Baptiste, et al. "Flamingo: a visual language model for few-shot learning." arXiv preprint arXiv:2204.14198 (2022).
> >
> > [2] Vedantam, Ramakrishna, C. Lawrence Zitnick, and Devi Parikh. "Cider: Consensus-based image description evaluation." Proceedings of the IEEE conference on computer vision and pattern recognition. 2015.
> >
> > [3] Zhao, Zihao, et al. "Calibrate before use: Improving few-shot performance of language models." International Conference on Machine Learning. PMLR, 2021

---

### Meta-Review · Area_Chair_ZAjo · 2022-08-26

**Recommendation:** Accept
**Confidence:** Certain

**Metareview:**

While the reviewers are divided, I agree with those with accept. The paper introduces an interesting alternative without training on videos and numbers useful for the community. The use of pretrained models in a clever way (without finetuning) is not a weakness but a contribution. A reviewer raised a concern about lacking analysis on how the proposed pipeline contributes to the few/zero-shot capability but it is already widely accepted that large-scale pretrained language models can do well in few/zero-shot settings with proper prompts. Also, the use of CLIP is just a way of extracting textual/categorical representation from the input video using a pretrained network and I believe the authors chose to use CLIP mainly because it is trained on a large-scale dataset with an open vocabulary. I think engineering this component is not the main focus of the paper and the lack of ablations on this should not discount the paper’s novelty.

**Award:**

No

---

### Decision · Program_Chairs · 2022-09-14

Accept